# Decoupling excitons from high-frequency vibrations in organic molecules

Pratyush Ghosh[1], Antonios M. Alvertis[2,3], Rituparno Chowdhury[1], Petri Murto[1,4], Alexander J. Gillett[1], Shengzhi Dong[5], Alexander J. Sneyd[1], Hwan-Hee Cho[1], Emrys W. Evans[1,6], Bartomeu Monserrat[1,7], Feng Li[5], Christoph Schnedermann[1], Hugo Bronstein[1,4], Richard H. Friend[1] & Akshay Rao[1✉]

The coupling of excitons in π-conjugated molecules to high-frequency vibrational modes, particularly carbon–carbon stretch modes (1,000–1,600 cm$^{-1}$) has been thought to be unavoidable[1,2]. These high-frequency modes accelerate non-radiative losses and limit the performance of light-emitting diodes, fluorescent biomarkers and photovoltaic devices. Here, by combining broadband impulsive vibrational spectroscopy, first-principles modelling and synthetic chemistry, we explore exciton–vibration coupling in a range of π-conjugated molecules. We uncover two design rules that decouple excitons from high-frequency vibrations. First, when the exciton wavefunction has a substantial charge-transfer character with spatially disjoint electron and hole densities, we find that high-frequency modes can be localized to either the donor or acceptor moiety, so that they do not significantly perturb the exciton energy or its spatial distribution. Second, it is possible to select materials such that the participating molecular orbitals have a symmetry-imposed non-bonding character and are, thus, decoupled from the high-frequency vibrational modes that modulate the π-bond order. We exemplify both these design rules by creating a series of spin radical systems that have very efficient near-infrared emission (680–800 nm) from charge-transfer excitons. We show that these systems have substantial coupling to vibrational modes only below 250 cm$^{-1}$, frequencies that are too low to allow fast non-radiative decay. This enables non-radiative decay rates to be suppressed by nearly two orders of magnitude in comparison to π-conjugated molecules with similar bandgaps. Our results show that losses due to coupling to high-frequency modes need not be a fundamental property of these systems.

In the limit of weak electronic coupling between the ground and excited electronic states, the rate of non-radiative recombination ($k_{nr}$) as a function of the energy gap $\Delta E$ between the excited and ground electronic states can be written as[1,3,4]

$$k_{nr} = \frac{C^2 (2\pi)^{1/2}}{\hbar (\hbar\omega\Delta E)^{1/2}} \exp\left[ -\frac{\Delta E}{\hbar\omega}\left\{\ln\left(\frac{\Delta E}{\sum_i \lambda_i}\right) - 1\right\}\right] \quad (1)$$

where $C$ is the effective electronic coupling matrix element and $\lambda_i$ corresponds to the reorganization energy associated with the driving modes that promote[5] non-radiative relaxation. Equation (1) formalizes the energy-gap law, which predicts an increase in non-radiative decay rate with a decreasing energy gap.

High-frequency molecular vibrations (1,000–1,600 cm$^{-1}$) are ubiquitous in π-conjugated organic molecules and are strongly coupled to electronic excited states where they directly modulate the

π-bond order[1]. This is particularly problematic when the energy gap is directly coupled to π-bonding alternations, such as the phenylene ring-stretching mode[6]. On the other hand, low-frequency modes of less than 500 cm$^{-1}$ are associated with high-mass displacement and are structurally delocalized in nature. These modes contribute less to fast non-radiative decay, which is captured by the $\omega$ term in equation (1). This term is the frequency of the promoting vibrational mode. Thus, exciton–vibration coupling to high-frequency modes, as generally observed for molecular systems[7–11], causes rapid non-radiative decay dynamics. The key question is, therefore, whether we can decouple excitons from high-frequency vibrations in organic molecules.

## Radicals as an efficient NIR emitter

We have selected a few examples of recently developed near-infrared (NIR) organic emitters that appear to violate the energy-gap law. We begin by focusing on an emerging family of spin-1/2 radical molecular

[1]Cavendish Laboratory, University of Cambridge, Cambridge, UK. [2]KBR, Inc., NASA Ames Research Center, Moffett Field, CA, USA. [3]Materials Sciences Division, Lawrence Berkeley National Laboratory, Berkeley, CA, USA. [4]Yusuf Hamied Department of Chemistry, University of Cambridge, Cambridge, UK. [5]State Key Laboratory of Supramolecular Structure and Materials, College of Chemistry, Jilin University, Changchun, China. [6]Department of Chemistry, Swansea University, Swansea, UK. [7]Department of Materials Science and Metallurgy, University of Cambridge, Cambridge, UK. ✉e-mail: ar525@cam.ac.uk

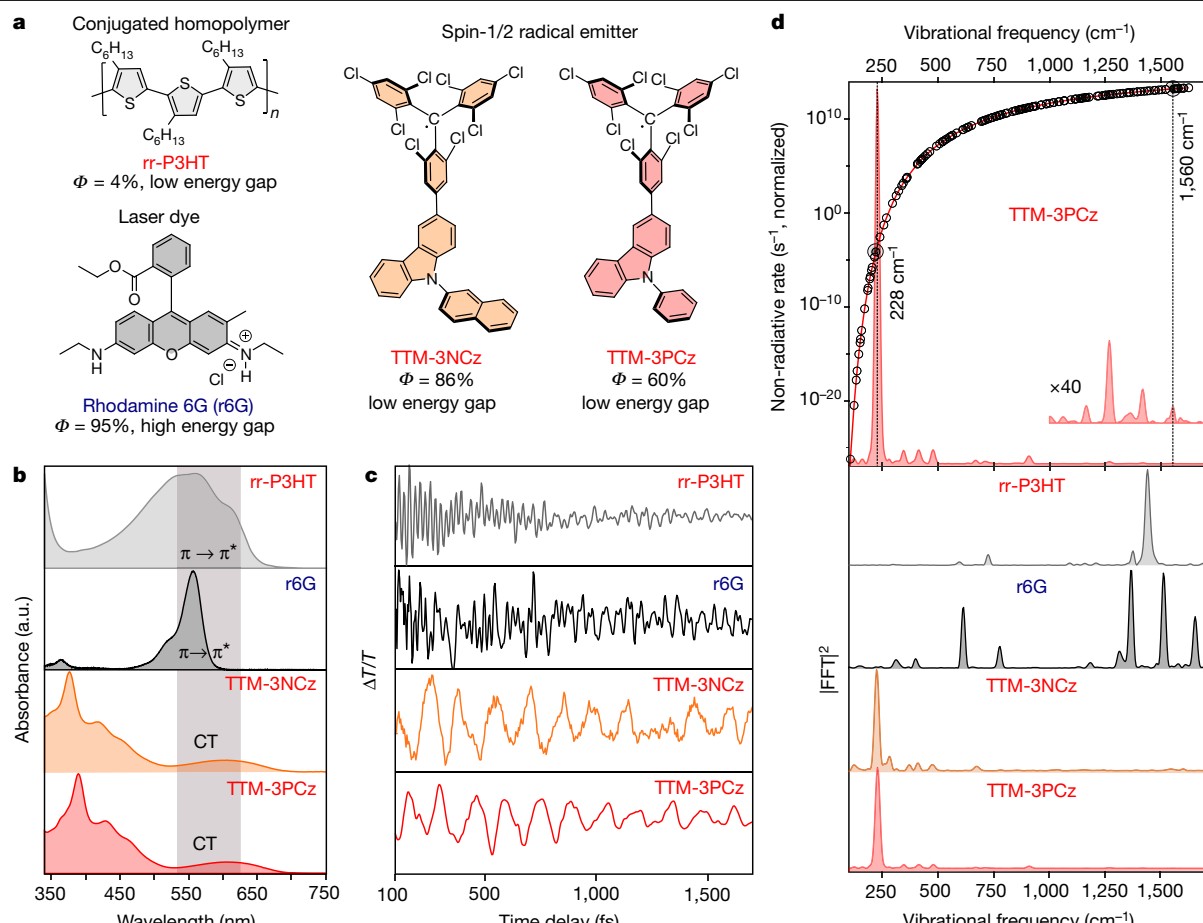

**Fig. 1 | Vibrational coherence as a probe of exciton–vibration coupling.**
**a**, Molecular structures of the conjugated polymer (rr-P3HT), a laser dye (rhodamine 6G or r6G), two conventional organic semiconductors and spin-1/2 radical emitters (TTM-3PCz and TTM-3NCz), which are both state-of-the-art deep-red/NIR emitters. The name of the molecule is highlighted in red if the corresponding emission maxima are above 650 nm. The maximum reported PLQEs, Φ, are indicated. Note that rhodamine 6G has a high PLQE despite strong coupling to high-frequency modes due to the high energy-gap emission.
**b**, Absorption spectra and the transition involved for impulsive excitation. The grey rectangular box indicates the spectral profile of the impulsive pump used to excite the samples. **c**, Vibrational coherence extracted from excited-state signal. **d** (bottom), The corresponding excited-state Raman spectra obtained after time-resolution correction for rr-P3HT (grey), rhodamine 6G (black), TTM-3NCz (orange) and TTM-3PCz (red). **d** (top), Theoretically calculated non-radiative decay rate (from equation (1) using DFT and TDDFT) plotted against the vibrational frequency for TTM-3PCz excitons assuming the corresponding vibrational mode as the main deactivation pathway. The black circles indicate the normal modes of the TTM-3PCz molecule. The probe windows for the vibrational coherence shown in **c** are as follows: rr-P3HT, 700–710 nm (stimulated emission)[35]; rhodamine 6G, 560–570 nm (stimulated emission); TTM-3NCz, 650–710 nm (excited-state photo-induced absorption plus stimulated emission) and TTM-3PCz: 660–670 nm (excited-state photo-induced absorption and stimulated emission convoluted). The further analysis in Supplementary Information sections 5 and 6 investigates the origin of the vibrational coherence for TTM-3PCz molecules, which confirms that high-frequency decoupled vibrational coupling corresponds to the transition involved in luminescence. All samples were measured in solution except for P3HT (Methods). a.u., arbitrary units.

semiconductors, as these have some of the highest values for the photoluminescence quantum efficiency (PLQE) and electroluminescence quantum efficiency (EQE$_{EL}$) reported for organic systems. These molecules consist of a doublet spin unit (TTM), which acts as an electron acceptor covalently linked to a donor unit, and an N-arylated carbazole moiety (TTM-3PCz and TTM-3NCz); see Fig. 1a. Both materials show strong luminescence from an intra-molecular charge-transfer exciton[12,13]. The lowest optical excitation for absorption and emission is the charge-transfer transition within the spin doublet manifold. With the correct tuning of the charge-transfer energetics and overlap, efficient emission can be achieved[14]. For these materials, excitation within the doublet manifold avoids access to higher multiplicity spin states and eliminates the problems due to the formation of triplet excitons in conventional closed-shell organic emitters. These systems have a very high NIR PLQE. For example, for TTM-3NCz, it is above 85% for the solid-state blends used in organic light-emitting diode host CBP for emission at 710 nm (ref. 12). The key question is how these systems can

achieve such a high luminescence yield in the NIR and, thus, seemingly violate the energy-gap law.

To answer this question, we compared these materials with a range of conventional organic molecules: regioregular poly(3-hexylthiophene) or rr-P3HT, a well-studied semiconductor homopolymer with a low PLQE of less than 5% at 680 nm, and the laser dye rhodamine 6G (r6G), which emits brightly at a relatively higher energy gap (PLQE = 94% at 550 nm).

## Impulsive vibrational spectroscopy

We probe the vibrational coupling in the excited electronic state of these organic molecules by employing resonant impulsive vibrational spectroscopy (IVS; Methods and Supplementary Information section 18)[15].

Figure 1b shows the absorption spectra of the organic molecules investigated as well as the spectral range of the ultrafast pump pulse used in our IVS studies (grey rectangle, 8.8 fs, centred at 575 nm).

For rr-P3HT and r6G, the pump pulse was resonant with a π → π* transition. By contrast, for TTM-3PCz and TTM-3NCz, the pump pulse was resonant with a charge-transfer transition, which corresponds to a doublet excitation ($D_0 → D_1$) from the 3PCz/3NCz-centred highest occupied molecular orbital (HOMO) to a TTM-centred singly occupied molecular orbital (SOMO)[12].

Following resonant impulsive excitation by the pump pulse, the early-time electronic population dynamics exhibits distinct oscillatory modulations across the entire visible probe region for all investigated molecules (see Extended Data Figs. 1c,d and 4a for wavelength-resolved analysis of TTM-3PCz and TTM-3NCz, respectively). Figure 1c displays the isolated excited-state vibrational coherences, and Fig. 1d (bottom) shows the time-resolution-corrected excited-state Raman spectrum of the corresponding time-domain data from Fig. 1c for each molecule (see Methods for details). We observe in rr-P3HT a pronounced vibrational mode at 1,441 cm⁻¹, which is due to the C=C ring-stretching mode in this conjugated polymer system[7]. Rhodamine 6G has a series of high-frequency modes (1,356, 1,504 and 1,647 cm⁻¹) corresponding to localized C–C and C=C stretching motions.

By contrast, the excited-state vibrational spectra of TTM-3PCz and TTM-3NCz have only one prominent mode (232 cm⁻¹), which is in the range of frequencies associated with torsional motions of the TTM to 3PCz/3NCz moiety (see Extended Data Fig. 3 for a complete analysis of low-frequency modes and theoretically calculated exciton–vibration coupling constants).

These results suggest that two different regimes for exciton–phonon coupling operate in the materials studied here. For the conjugated homopolymer (rr-P3HT) and laser dye (rhodamine 6G), the photo-excited transition leads to the formation of excitons coupled to high-frequency C–C and C=C stretching modes, as is conventionally expected for organic systems. However, the lowest lying excitons of TTM-3PCz and TTM-3NCz, which are associated with the charge-transfer $D_0 → D_1$ transition, are decoupled from these high-frequency modes.

The effect of this vibrational decoupling on the non-radiative loss is dramatic. This is illustrated in the top panel of Fig. 1d, which shows the non-radiative decay rate along each normal mode coordinate calculated using density functional theory (DFT). It can be seen that the high-frequency modes lead to significantly faster rates of non-radiative recombination. For instance, taking $\Delta E = 14,437$ cm⁻¹ (1.79 eV) and a reorganization energy of 1,105 cm⁻¹ (0.137 eV) (Supplementary Information section 12), for TTM-3PCz, a representative high-frequency 1,560 cm⁻¹ mode leads to a non-radiative rate approximately 10¹⁵ times faster that of the low-frequency 230 cm⁻¹ mode. In typical organic systems with a low bandgap, this would lead to rapid non-radiative losses. However, the key point here is that TTM-3PCz and TTM-3NCz show no coupling to these high-frequency modes.

## Band-selective impulsive excitation

Having probed the charge-transfer type $D_0 → D_1$ transitions in radical molecules, we now turn our attention to the higher $D_0 → D_2$ transition, which does not involve charge-transfer excitons[12,16]. Studying this transition, therefore, allowed us to compare the vibrational coupling between the charge-transfer and non-charge-transfer transitions of the same molecule. Here, we focus on the novel radical molecule TTM-TPA, which has a donor–acceptor structural motif like those of TTM-3PCz and TTM-3NCz but has a triphenylamine (TPA) group as electron donor instead of the N-aryl carbazole (PCz/NCz) group (Fig. 2a). As shown in Fig. 2a and like TTM-3PCz (ref. 12), the lowest-energy electronic transition ($D_0 → D_1$) in TTM-TPA corresponds to a charge-transfer excitation from the TPA-centred HOMO (donor) to the TTM-centred SOMO (acceptor), as revealed by time-dependent DFT (TDDFT) calculations (Extended Data Table 2). The second lowest-energy transition ($D_0 → D_2$) involves frontier molecular orbitals sitting predominantly on the TTM

part (HOMO-2 to SOMO), corresponding to spatially overlapping orbitals, which is consistent with similar derivations[12,16]. As illustrated in Fig. 2b, the charge-transfer character of the lowest-energy absorption band (approximately 700 nm, $D_0 → D_1$) shows the expected solvatochromic redshift, whereas the higher energy absorption band (approximately 500 nm, $D_0 → D_2$) is barely affected by the solvent polarity. As TPA has a higher-lying HOMO compared to 3PCz (ref. 17), the charge-transfer transition is redshifted while it maintains a nearly similar energy for the local exciton transition in TTM-TPA (Fig. 2b). This greater energy separation between the charge-transfer and non-charge-transfer transitions allows us to compare their vibrational coupling more cleanly than would be possible in TTM-3PCz. We excited the charge-transfer state with a pump pulse centred at 725 nm (pulse $P_1$, 12 fs, Fig. 2b), whereas the local exciton state was excited with a pump pulse centred at 575 nm (pulse $P_2$, 15 fs, Fig. 2b).

Photo-excitation of TTM-TPA into $D_1$ with pulse $P_1$ yielded vibrational coherences like those of the previously observed charge-transfer excitons of TTM-3PCz and TTM-3NCz. Here, the excited-state impulsive vibrational spectrum is again dominated by low-frequency modes (228 cm⁻¹) with a minor contribution from high-frequency modes in the range 1,100–1,650 cm⁻¹ (magenta, Fig. 2c). Photo-excitation into the non-charge-transfer exciton state through pulse $P_2$ populated the $D_2$ state, which rapidly cooled to the $D_1$ state with a time constant of 670 fs (see Extended Data Fig. 5 for the electronic and vibrational dynamics of $D_2 → D_1$ cooling). Figure 2c (purple) shows the corresponding vibrational spectrum obtained directly after photo-excitation into $D_2$, which exhibits significantly enhanced coupling to high-frequency modes at 1,272, 1,520 and 1,565 cm⁻¹, in stark contrast to the spectrum obtained for $D_1$ (Fig. 2c, magenta; see Extended Data Fig. 6 for a wavelength-resolved analysis). Taken together, this selective photo-excitation reveals that the charge-transfer ($D_1$) and non-charge-transfer exciton ($D_2$) states exhibit large differences in their coupling to the vibrational modes, even within the same molecule.

## First-principles modelling

We performed first-principles spin-unrestricted DFT and TDDFT calculations[18–20] to quantify the exciton–vibration coupling using the Huang–Rhys factor ($S_{ev}$) for the $D_0 → D_1$ (charge-transfer) and $D_0 → D_2$ (non-charge-transfer) electronic transitions:

$$S_{ev(i)}(k) = \left( \frac{E_{ex(i)}^{+\delta u(k)} - E_{ex(i)}^{-\delta u(k)}}{2\delta u(k)\hbar\omega} \right)^2, \quad (i = D_0 → D_1/D_0 → D_2). \quad (2)$$

$E_{ex(i)}^{+\delta u(k)}$ and $E_{ex(i)}^{-\delta u(k)}$ are the excitation energies for an electronic transition ($i$) upon displacing the equilibrium geometry by a small dimensionless quantity ($+\delta u$, $-\delta u$) along the $k$th normal mode with frequency $\omega$, in the harmonic limit.

We computed $S_{ev}(\omega)$ for the normal modes of TTM-TPA associated with both the $D_0 → D_1$ (charge-transfer) and $D_0 → D_2$ (non-charge-transfer) excitations. The results are shown in Fig. 2d for the key high-frequency vibrational modes in the experimental data, namely 1,273, 1,522, 1,561 and 1,572 cm⁻¹. The calculations show that for all high-frequency modes, the vibrational coupling is significantly reduced for the charge-transfer exciton (Fig. 2d, magenta) compared to the non-charge transfer exciton (Fig. 2d, purple), in line with the experimental observations

To better understand how these vibrational modes affect the electronic structure of TTM-TPA, we computed the exciton wavefunction ($\rho$) as well as the change in the wavefunction due to displacement along a normal mode, $\{\Delta\rho\}_\omega$. Figure 2f,g shows these differential wavefunction plots $\{\Delta\rho\}_\omega$, for the normal mode with a frequency of 1,561 cm⁻¹, which is associated with C–C and C=C stretching vibrations localized primarily on the TTM moiety (Fig. 2e). This mode is strongly present in the experimental data (1,565 cm⁻¹, Fig. 2c) and also

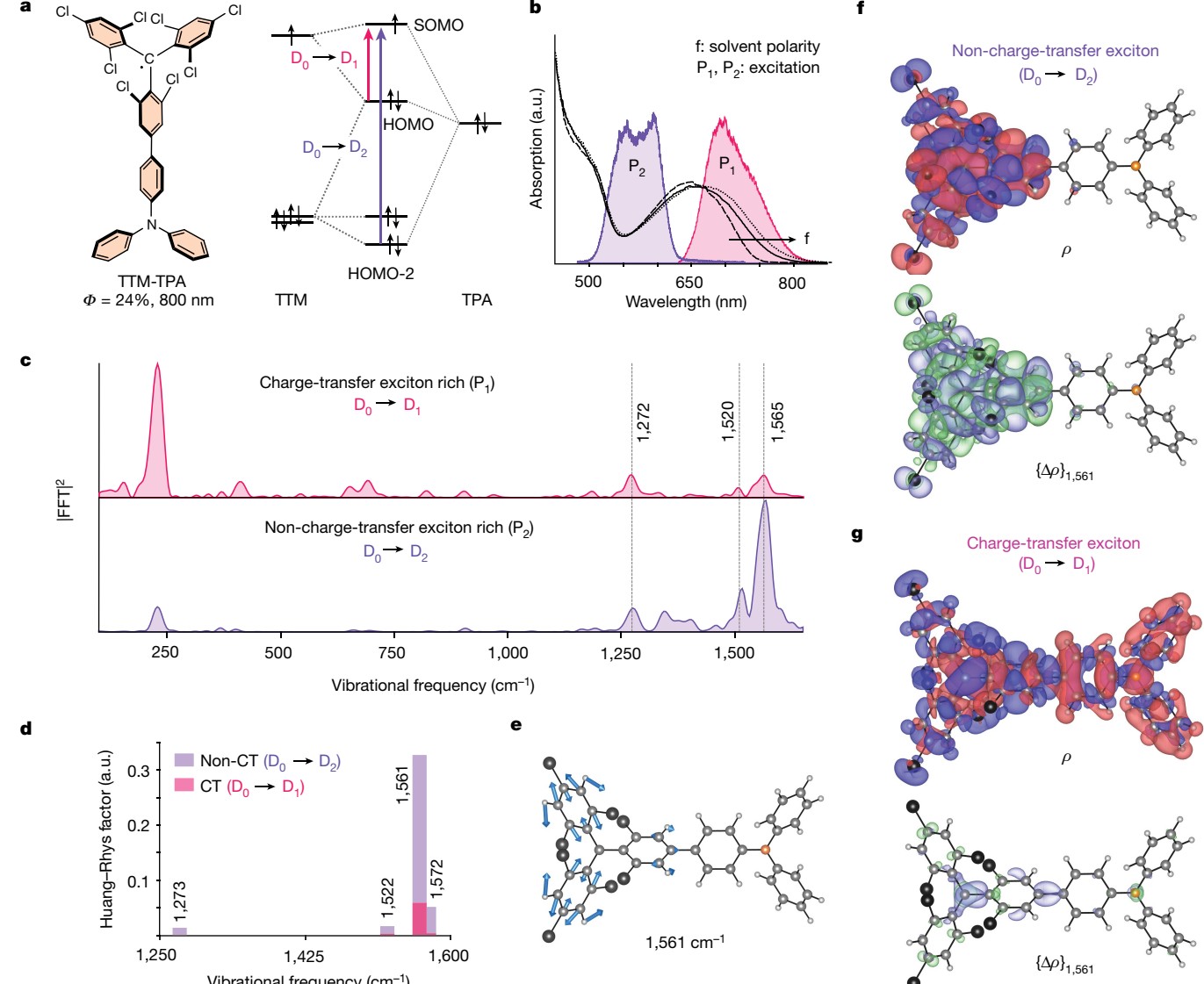

**Fig. 2 | Isolation of the vibrational coupling for the charge-transfer and non-charge-transfer excitons of an efficient doublet emitter. a**, Chemical structure of the novel NIR emitter TTM-TPA and its molecular orbital diagram highlighting the two lowest-energy transitions used in the band-selective excitation indicated by the purple and magenta arrows. The energy levels in the molecular orbital diagram are not to scale. **b**, Steady-state absorption spectra of TTM-TPA in different solvents with variable polarity (f: cyclohexane → $CHCl_3$ → tetrahydrofuran). The magenta and purple areas indicate the spectral profile of the impulsive pumps ($P_1$ and $P_2$) used to excite the different bands. **c**, Excited-state vibrational spectra obtained from vibrational coherence generated at early timescales (100–1,250 fs) upon exciting with the magenta and purple impulsive pumps. **d**, Theoretically calculated exciton–vibration coupling parameter, the so-called Huang–Rhys factor ($S_{ev}(k)$), for the $D_0 \rightarrow D_1$ and $D_0 \rightarrow D_2$ electronic transitions of TTM-TPA in the high-frequency regime for the experimentally obtained modes. CT, charge transfer. **e**, Vector displacement diagram of the high-frequency breathing mode with frequency 1,561 $cm^{-1}$ plotted on the optimized geometry of TTM-TPA. **f,g** (top), Exciton wavefunction (transition density) $\rho$ for the $D_2$ (non-charge-transfer exciton) transition (**f**) and the $D_1$ (charge-transfer exciton) transition (**g**). **f,g** (bottom), differential exciton wavefunction (transition density) upon displacement along the 1,561 $cm^{-1}$ mode $\{\Delta\rho\}_{1,561\,cm^{-1}}$, plotted for the $D_2$ (non-charge-transfer exciton) transition (**f**) and the $D_1$ (charge-transfer exciton) transition (**g**).

dominates the theoretically calculated exciton–vibration coupling plot (Fig. 2d).

TDDFT calculations reveal that excitation to $D_2$ localizes the wavefunction onto TTM ($\rho$, Fig. 2f, top), as expected for a local non-charge-transfer excitonic state[16]. Figure 2f (bottom) shows how the exciton density on the molecule varies owing to perturbations of the molecular geometry along the 1,561 $cm^{-1}$ mode, which is represented by $\{\Delta\rho\}_{1,561}$. Displacement along this normal mode leads to large changes in the $D_2$ exciton wavefunction, indicating strong coupling of the high-frequency vibrational modes to the non-charge-transfer exciton wavefunction. We then compare this to the exciton density arising from $D_0 \rightarrow D_1$ (Fig. 2g, top), which leads to a delocalized wavefunction over the whole molecule with disjoint electron and hole densities. Critically, the exciton density upon perturbation along the 1,561 $cm^{-1}$ normal mode ($\{\Delta\rho\}_{1,561}$, Fig. 2g, bottom) shows very little change for the $D_1$ exciton, in marked contrast to the results for the $D_2$ exciton. This shows the strong suppression of exciton–vibration coupling for the charge-transfer-type $D_1$ exciton. Extended Data Fig. 7 presents similar results for all the other experimentally obtained high-frequency vibrational modes.

To get a complete mode-resolved picture, we also calculated the parameter $\varphi_{lf}^{hf}$, which is defined as the ratio of the vibrational reorganization energy[21] ($\lambda_v$) associated with all high-frequency normal modes (1,000–2,000 $cm^{-1}$) to the low-frequency modes (100–1,000 $cm^{-1}$) for a particular electronic transition:

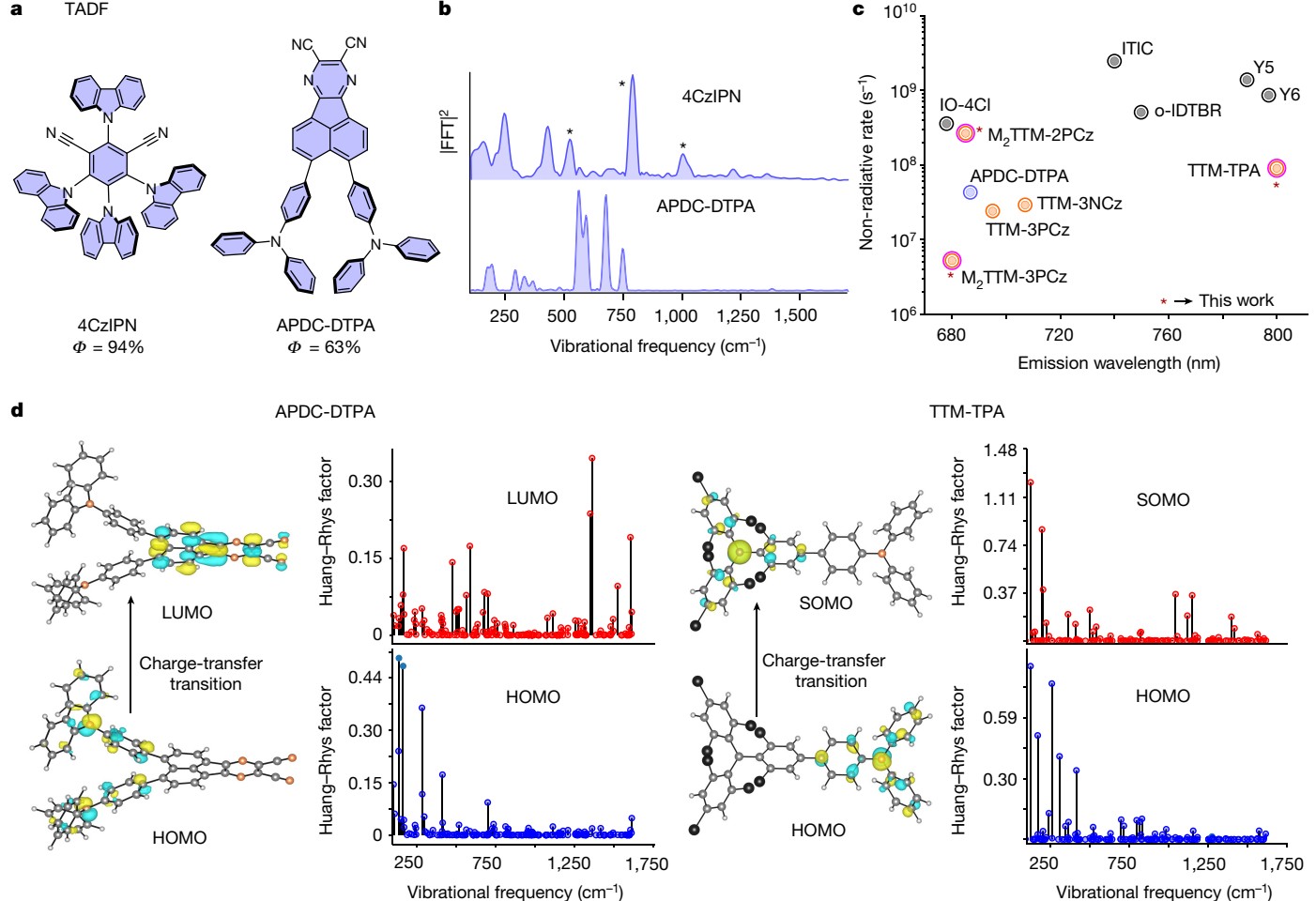

**Fig. 3 | Vibrational coupling to the organic excitons with variable charge transfer. a**, Molecular structures of the TADF molecules studied (4CzIPN and APDC-DTPA) **b**, Time-resolution-corrected excited-state Raman spectra of the TADF (4CzIPN and APDC-DTPA). Asterisks indicate the solvent mode. See Extended Data Figs. 1a and 4b for a wavelength-resolved analysis of APDC-DTPA and 4CzIPN. The three-pulse IVS experiment on 4CzIPN solution is detailed in Supplementary Information section 7. **c**, Experimentally obtained non-radiative rates of the studied low energy-gap molecules. Orange circles represent radical emitters. The blue circle represents TADF. The grey circles represent non-fullerene acceptors[36] (IO-4Cl, ITIC, o-IDTBR, Y5 and Y6). **d**, Vibrational coupling to the frontier molecular orbitals of APDC-DTPA (TADF)

and TTM-3PCz (radical) obtained from first-principles DFT. The hole-accepting orbitals of both TADF (HOMO of APDC-DTPA) and the radical (HOMO of TTM-TPA) are localized on the central N atom with a non-bonding character, which is reflected in the lower coupling to the high-frequency phenylic ring-stretching modes. The electron-accepting level of the radical (SOMO of TTM-TPA) has a non-bonding character and suppressed high-frequency coupling with respect to its HOMO, whereas for the TADF structures, the hole-accepting level (HOMO of APDC-DTPA) makes a significant orbital contribution in the vicinity of the planar π bonds and shows reasonable high-frequency coupling.

$$\varphi_{lf}^{hf} = \frac{\lambda_v^{hf}}{\lambda_v^{lf}} \text{ where } \lambda_v^{hf} = \sum_{\omega_k=1,000 \text{ cm}^{-1}}^{2,000 \text{ cm}^{-1}} \hbar\omega_k S_{ev}(k)$$

$$\text{and } \lambda_v^{lf} = \sum_{\omega_k=100 \text{ cm}^{-1}}^{1,000 \text{ cm}^{-1}} \hbar\omega_k S_{ev}(k). \tag{3}$$

As displayed in Extended Data Fig. 8a,b, $\varphi_{lf}^{hf}$ for the non-charge-transfer-type $D_2$ state is 2.4 times higher than for the charge-transfer-type $D_1$ state, in agreement with the experimental results in Fig. 2d.

## Thermally activated delayed fluorescence

We next examined how other low-bandgap organic systems, especially those with a variable charge-transfer character in the exciton coupling to vibrations (Fig. 3a). We selected APDC-DTPA (refs. 22–24) as an example of a highly efficient NIR-emitting thermally activated delayed fluorescence (TADF) system (PLQE = 63% at 687 nm)[22]. Here,

the electronic excitation promotes an electron from the HOMO centred at TPA to the lowest unoccupied molecular orbital (LUMO) at an acenaphthene-based acceptor core (APDC). We also studied a classic green-emitting TADF system, 4CzIPN, which has a higher energy gap (refs. 25–29; PLQE = 94% at 550 nm). The key design feature of TADF systems, as first developed by Adachi and co-workers[28], is the introduction of a donor–acceptor character such that the charge-transfer exciton has a spatially reduced electron–hole overlap that reduces the singlet–triplet exchange energy.

As shown in Fig. 3b, the excited-state vibrational spectrum of APDC-DTPA exhibits vibrational activity only in the lower-frequency regime (183, 290, 324, 557, 587, 678 and 735 cm⁻¹), which is associated with more delocalized torsional modes in the system. Similarly, 4CzIPN shows strong coupling to low-frequency torsional modes (157, 244, 429, 521 and 562 cm⁻¹) with a nominal contribution from high-frequency modes.

Once again, we observed that electronic transitions featuring a strong charge-transfer character and non-planar molecular geometry, which lead to spatially separated and disjoint HOMO/SOMO or the LUMO,

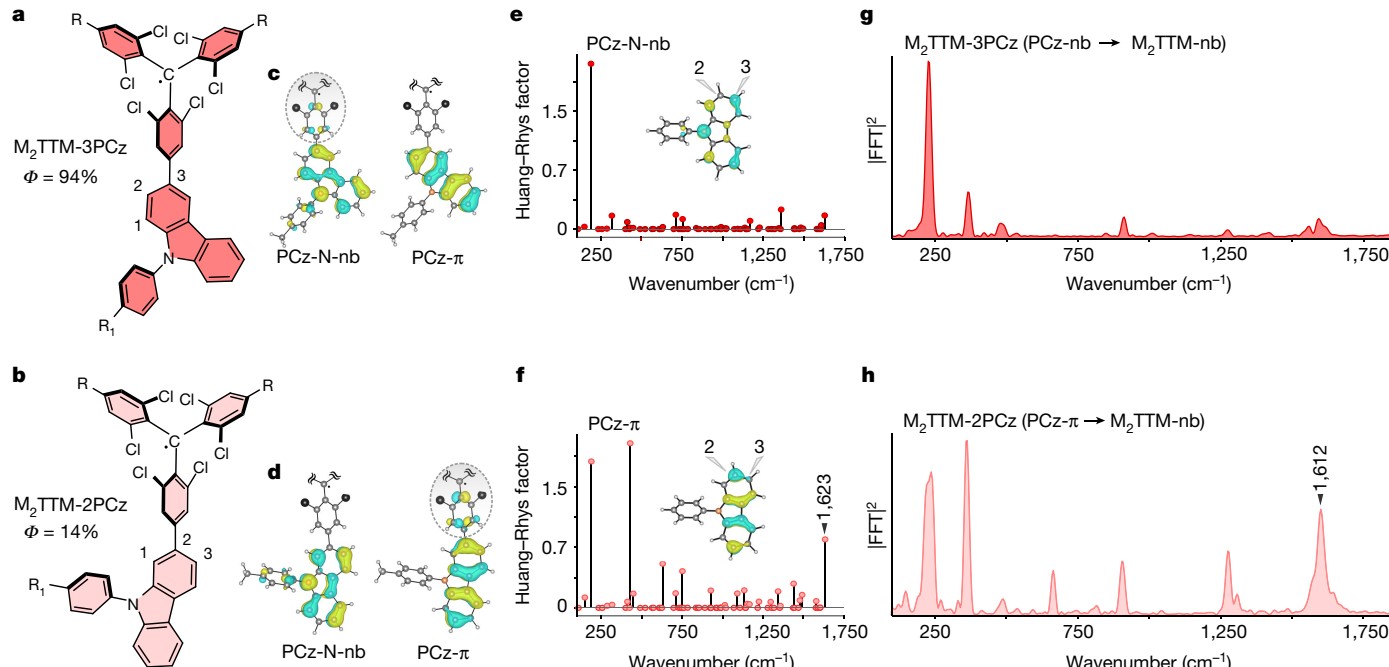

**Fig. 4 | Optimizing the photoluminescence efficiency by tuning the non-bonding character in the HOMO or hole-accepting level. a,b,** Chemical structure of the regio-isomers of dimesitylated-TTM-carbazole system M₂TTM-3PCz (**a**) and M₂TTM-2PCz (**b**) with solution PLQE. R stands for the mesityl group and R₁ represents –C₆H₁₃. **c,d,** Two HOMOs having N-non-bonding and π character and their alternative pattern of delocalization in the phenyl group of the M₂TTM moiety upon changing the linking position from 3 (**c**) to 2 (**d**). The grey dotted circle represents the delocalization of the molecular orbital from the PCz group to the adjacent phenyl ring of the M₂TTM radical core. **e,f,** Vibrational coupling parameters for the N-non-bonding (**e**) and π (**f**) molecular orbitals of phenyl-carbazole (PCz). **g,h,** Time-resolution-corrected excited-state Raman spectra of M₂TTM-3PCz (**g**) and M₂TTM-2PCz (**h**). The selected probe windows are the blue edges of the photo-induced absorption (680–700 nm for M₂TTM-3PCz and 680–700 nm for M₂TTM-2PCz; see Extended Data Fig. 9 for a wavelength-resolved analysis).

as seen for APDC-DTPA, 4CzIPN, TTM-3PCz, TTM-3NCz and TTM-TPA, give rise to excitons that do not couple to high-frequency modes.

Figure 3c summarizes the non-radiative decay rates for all the deep-red/NIR-emitting molecules studied here, which were based on radiative lifetime and PLQE measurements. These measurements directly show that the suppression of coupling to high-frequency modes, as measured by IVS (Figs. 1d, 2c and 3b), results in a lower non-radiative decay rate in doublet and TADF systems. This contrasts with non-fullerene acceptor systems, which have higher rates of non-radiative decay and which we found have strong coupling to high-frequency modes, as presented in Extended Data Fig. 2 and Supplementary Information section 21. By contrast, the TADF and radical emitters show greatly suppressed non-radiative rates, matching the lack of coupling to high-frequency modes observed by IVS.

This is also in agreement with our calculations, which show a supressed contribution of the high-frequency normal modes to the vibrational reorganization energy for the charge-transfer excitons studied here (TTM-3PCz and APDC-DTPA), in comparison to the non-charge-transfer excitons (TTM and pentacene), in agreement with non-adiabatic calculations[13] (Extended Data Fig. 8c), which is further supported experimentally by solvent polarity-dependent IVS measurements (Supplementary Information section 19). Taken together, the calculations support the experimental observation of suppressed high-frequency vibrational activity of the excited electronic state for charge-transfer excitations.

Intuitively, a general proposition to understand these results can be understood as follows. The charge-transfer excitation in non-planar molecules (radicals and TADF) provides spatially separated electrons (HOMOs/SOMOs) and holes (LUMOs) across the molecular backbone (Figs. 2g, 3d). Simultaneous changes to both the electron and hole wavefunctions due to highly localized high-frequency carbon–carbon

stretching motion, therefore, result in a smaller effect compared to planar excitonic systems, which exhibit strongly overlapping HOMOs and LUMOs with high electronic densities in the vicinity of these high-frequency nuclear oscillations. We note that although the non-fullerene acceptor systems have a donor–acceptor structural motif, due to their coplanar geometry and strong electronic conjugation through the fused rings, the HOMO and LUMO strongly overlap in space, unlike the radicals and TADF, so that the dipole oscillator strength of the lowest-energy transition in these materials is very high, as required for their use in photovoltaics.

## Non-bonding-type electron and hole levels

Comparing the measured impulsive vibrational spectra of the radical emitters with the TADF molecules (Figs. 1d and 3b), it can be seen that although the TADF molecules do not couple to vibrations of more than 1,000 cm⁻¹, the radicals do not couple to modes of more than 240 cm⁻¹. The radical emitters also display lower rates of non-radiative recombination than the TADF systems, as shown in Fig. 3c and, thus, go beyond what these TADF systems can achieve in terms of supressing non-radiative recombination. This suggests that something else is also supressing the coupling to high-frequency modes in radical systems in comparison to TADF systems.

Figure 3d shows the results of first-principles calculations for the vibrational coupling to the hole and electron levels in APDC-DTPA (TADF molecule) and TTM-TPA (radical molecules). In both TADF and radical systems, TPA and N-aryl-carbazole (Cz) are widely adopted as donors[12,17,30]. These donor moieties have nitrogen $p_z$ centred non-bonding-type HOMO. As can be seen in Fig. 3d, these levels are not strongly coupled to high-frequency phenyl ring-stretching vibra-tions. The degree of further localization of the non-bonding-type

HOMO on the nitrogen atom depends on the non-planarity of the nitrogen $p_z$ orbital to the adjacent π systems imposed by steric hindrance[31] (Supplementary Information section 14). It can be seen that for the electron level of the TADF molecule, the LUMO does show coupling to high-frequency vibrations, but for the radical systems, the electron-accepting SOMO level, localized on the TTM moiety's central $sp^2$ carbon atom, has a non-bonding character[12] and does not show strong coupling to high-frequency vibrational modes. This implies that both the electron and hole levels for the radical systems have a non-bonding character, whereas only the hole levels have this non-bonding character in TADF systems (Fig. 3d). This may explain the weaker coupling to high-frequency vibrations (Figs. 1d and 3b) and the lower non-radiative recombination rate in the radical systems compared to the TADF systems.

To test this hypothesis, we designed two radical molecules to tune the participation of the non-bonding character in the emitting electronic transition. As can be seen in Fig. 4a,b, $M_2$TTM-3PCz and $M_2$TTM-2PCz are two regio-isomers of the dimesitylated-TTM linked through either the 3 or 2 positions of phenylcarbazole (PCz). This regioselective linking between donor and acceptor leads to dramatic changes in the photoluminescence efficiencies (for $M_2$TTM-3PCz, PLQE = 92% with a photoluminescence lifetime of 15.2 ns, whereas for $M_2$TTM-2PCz, PLQE = 14% with a photoluminescence lifetime of 3.2 ns) and a very large difference in the non-radiative rate (Extended Data Fig. 9d). As can be visualized in Fig. 4c,d, from an electronic structure point of view, the molecules are differentiated because either the N-non-bonding type HOMO (for $M_2$TTM-3PCz) or the carbazole-π-type HOMO-1 (for $M_2$TTM-2PCz) are delocalized onto the adjacent phenyl ring of the $M_2$TTM moiety. On the basis of our hypothesis of the importance of the non-bonding character in suppressing coupling to high-frequency vibrations, we would predict that $M_2$TTM-3PCz should show reduced coupling to high-frequency modes in comparison to $M_2$TTM-2PCz (Fig. 4e–f). This prediction is verified in the impulsive vibrational spectra in Fig. 4g–h, as $M_2$TTM-3PCz has stronger coupling to the carbazole ring-stretching node at 1,612 cm$^{-1}$ because the hole-accepting level has a carbazole-π character (see Extended Data Fig. 9c for wavelength-resolved data). This tuning of the coupling to high-frequency modes through the participation of non-bonding levels in the electronic transitions allowed us to achieve an even lower non-radiative rate (Fig. 3c) and near-unity PLQE at 680 nm. This shows that combining a charge-transfer character with non-bonding orbitals is the key to decoupling excitons from higher-frequency vibrations (over 250 cm$^{-1}$) and that a charge-transfer character alone is not sufficient.

## Conclusions and outlook

Taken together, our experiments and calculations provide a mechanistic picture for how to decouple excitons from vibrational modes in organic systems. If an exciton wavefunction has a substantial charge-transfer character and the electron and hole wavefunctions are spatially separated, localized high-frequency modes (over 1,000 cm$^{-1}$) do not significantly perturb its energy or its spatial distribution (experimental data in Figs. 1d and 2c and calculations in Fig. 2d,f,g). As depicted in Figs. 2 and 3, spatially separated disjoint hole (HOMO) and electron (LUMO) pairs can be generated by having a non-coplanar electron-rich (donor) and an electron-deficient (acceptor) moiety in a molecule, although this comes at the expense of a lower radiative rate. The selection of moieties with non-bonding electronic levels, such as the hole-accepting TPA, arylated carbazole (HOMO) and the electron-accepting TTM-donor radicals (SOMO), further decouples the exciton from high-frequency vibrations (over 250 cm$^{-1}$). Consequently, non-radiative decay channels, which are dominated by high-frequency modes in organic molecules, can be efficiently suppressed.

This mechanism explains the high luminescence efficiency of the low-bandgap TTM-donor-based radical molecules as well as some low-bandgap TADF systems, for which the charge-transfer excitations are the lowest excited state. Our results also explain the apparent contradiction in the performance of these materials, for which the charge-transfer character of the electronic transition leads to a reduced oscillator strength and reduced radiative rate (Extended Data Table 1), which would normally be associated with a lower luminescence efficiency[32,33]. However, the suppression of non-radiative decay pathways due to the charge-transfer character of the excitations and non-bonding nature of the levels, as demonstrated here, overcomes this and enables high luminescence efficiency from these states.

This has important implications for the design of organic emitters for organic light-emitting diodes and NIR fluorescent markers for biological applications. The proposed design principles also open up new possibilities for organic photovoltaics, by allowing efficient radiative recombination in organic photovoltaics (such as achieved with metal-halide perovskites or GaAs solar cells) to boost the open-circuit voltage, the major outstanding challenge in the field[2,34]. This could enable device efficiencies well above 20% in future organic photovoltaics.

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

# Methods

## Materials

TTM-3PCz and TTM-3NCz were synthesized by the Suzuki coupling reaction as reported earlier[12]. The synthetic route of the novel radical TTM-TPA is extensively discussed in Supplementary Information section 1. $M_2$TTM-3PCz was synthesized following the recently reported Suzuki coupling and radical conversion procedures, and the novel radical $M_2$TTM-2PCz was prepared by the same procedures, which are discussed in Supplementary Information section 17. APDC-DTPA, 4CzIPN, rr-P3HT, rhodamine 6G, pentacene, ITIC, IO-4Cl, o-IDTBR, Y5, Y6 and Y7 were obtained from Lumtec, Ossila and Merck. The impulsive measurements of all samples were done in solution except for rr-P3HT. To prevent rr-P3HT from burning on the cuvette wall (in solution), a spin-coated thin-film measurement was done. In solution, it can show PLQE = 33% with a blueshifted emission compared to films. The higher PLQE in solution can be ascribed to an avoidance of interchain-state formation and a high gap emission.

## Impulsive vibrational spectroscopy

In IVS, an ultrafast pump pulse (sub-15 fs) resonant with the optical gap impulsively generates vibrational coherence in the photo-excited state of a material, which evolves in time according to the underlying excited-state potential energy surface. The impulsive response of the system was recorded by a time-delayed probe pulse that was spectrally tuned to probe excited-state resonances. The so-obtained vibrational coherence manifested as oscillatory modulations superimposed on top of the sample's transient population dynamics and provided direct access to the excited-state Raman spectrum of the material with a Fourier transformation. The IVS experiments were performed with a home-built set-up[37] seeded by a commercially available Yb:KGW amplifier laser (PHAROS, Light Conversion, 1,030 nm, 38 kHz and 15 W). A chirped white light continuum (WLC) spanning from 530 to 950 nm was used as a probe pulse. This was generated by focusing a part of the fundamental beam onto a 3 mm YAG crystal and collimating after it. The impulsive pump pulses were generated by a non-collinear optical parametric amplifier (NOPA), as reported previously[38]. The second- (515 nm) and third-harmonic (343 nm) pulses required to pump the NOPAs were generated with an automatic harmonic generator (HIRO, Light Conversion). The impulsive pump (experiments reported in Fig. 2) and $P_2$ pulse for the band-selective experiment were generated with a NOPA seeded by 1030-WLC and amplified by the third harmonic (343 nm). The $P_1$ pulse for the band-selective experiment was generated with a NOPA seeded by 1030-WLC and amplified by the second harmonic (515 nm). Pump pulses were compressed using a pair of chirped mirrors in combination with wedge prisms (Layertec). The spatio-temporal profile of the pulses was measured with a second-harmonic generation frequency-resolved optical gating (Supplementary Information section 3 and Extended Data Fig. 6). A chopper wheel in the pump beam path modulated the pump beam at 9 kHz to generate differential transmission spectra. The pump–probe delay was set by a computer-controlled piezoelectric translation stage (PhysikInstrumente) with a step size of 4 fs. The pump and probe polarizations were parallel. The transmitted probe was recorded by a grating spectrometer equipped with a Si line camera (Entwicklungsbüro Stresing) operating at 38 kHz with a 550 nm blazed grating. Solution samples were measured in a flow cell cuvette with an ultrathin wall aperture (Starna, Far UV Quartz, path length of 0.2 mm). Pulse compression was performed after a quartz coverslip (170 μm) was placed in the beam path of the frequency-resolved optical gating to compensate for the dispersion produced by the cuvette wall.

## Time-domain vibrational data analysis

After correcting for the chirp and subtracting the background, the kinetic traces for each probe wavelength were truncated to exclude time delays of less than 100 fs to prevent contamination from coherent artefacts. We subsequently extracted the residual oscillations from the convoluted kinetic traces after globally fitting the electronic dynamics by a sum of two exponential decaying functions with an offset over the whole spectral range. A series of signal-processing techniques were employed to convert the oscillatory time-domain signals to the frequency domain, including apodization (Kaiser–Bessel window, $\beta = 1$), zero-padding and a fast Fourier transformation (FFT). Before we produced the intensity spectra, the |FFT| amplitude was multiplied by a frequency-dependent scaling function to remove time-resolution artefacts (the time-resolution correction method is described in detail in Supplementary Information section 3.2).

## Computational methods

To study the ground state properties of the different molecules, we performed DFT calculations, employing the B3LYP hybrid functional and cc-pVDZ basis set as implemented within the software NWChem (ref. 39). For the open-shell systems discussed in this work, we performed spin-unrestricted DFT calculations, setting the multiplicity to two (doublet state). To compute the vibrational properties and the effect of vibrations on excited states, we coupled our molecular DFT calculations to finite displacement methods[20]. Excited-state properties were computed by TDDFT on top of the previously calculated DFT ground states using the B3LYP exchange-correlation functional and the same basis set as above. We verified for each of the studied open-shell systems that the computed ground and excited states did not suffer from spin contamination[40].

## Data availability

The data underlying all figures in the main text are publicly available from the University of Cambridge repository https://doi.org/10.17863/CAM.105569 (ref. 41).

## Code availability

The code for analysing the data from the IVS experiments used in the manuscript is accessible through open access at https://doi.org/10.17863/CAM.105569 (ref. 41).

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

**Acknowledgements** We thank D. Beljonne and T. J. H. Hele for valuable discussions. This project has received funding from the European Research Council under the European Union's Horizon 2020 research and innovation programme (Grant Agreement No. 101020167 (SCORS) to R.H.F. and Grant Agreement No. 758826 (SOLARX) to A.R.). This work has received funding from the Engineering and Physical Sciences Research Council (UK). R.H.F. acknowledges support from the Simons Foundation (Grant No. 601946). P.G. thanks the Cambridge Trust and the George and Lilian Schiff Foundation for a PhD scholarship and St John's College, Cambridge, for additional support. H.B. acknowledges EPSRC (grant no EP/S003126/1). P.M. has received funding from Marie Skłodowska-Curie Actions (Grant Agreement No. 891167). H.C. acknowledges the George and Lilian Schiff Foundation for PhD studentship funding. P.M. and H.C. also acknowledge the European Research Council for the European Union's Horizon 2020 research and innovation programme (grant agreement no. 101020167) for funding. S.D. and F.L. are grateful for receiving financial support from the National Natural Science Foundation of China (Grant No. 51925303). R.C. thanks the European Union's Horizon 2020 project for funding under its research and innovation programme through Marie Skłodowska-Curie Actions (Grant Agreement No. 859752, HEL4CHIROLED). B.M. acknowledges support from a Future Leaders Fellowship from UK Research and Innovation (UKRI; Grant No. MR/V023926/1), from the Gianna Angelopoulos Programme for Science, Technology, and Innovation, and from the Winton Programme for the Physics of Sustainability. The calculations in this work were performed using resources provided by the Cambridge Tier-2 system operated by the University of Cambridge Research Computing Service and funded by the Engineering and Physical Sciences Research Council (Grant No.

EP/P020259/1). A.J.G. thanks the Leverhulme Trust for an Early Career Fellowship (ECF-2022-445). This work was funded by the UKRI. For the purpose of open access, the authors have applied a Creative Commons Attribution (CC BY) licence to any Author Accepted Manuscript version arising.

**Author contributions** A.R. conceived the project. P.G. developed the project, designed and built the experiments, and performed the resonant IVS and transient absorption spectroscopy measurements and quantum modelling. P.G. analysed the vibrational spectroscopy data with input from C.S. P.G. performed the quantum chemical calculations with input from A.M.A. and B.M. P.M. synthesized and characterized the dimesitylated-TTM-carbazole isomers under the supervision of H.B. P.G. and H.C. prepared the samples for the experiments. R.C. determined the PL characterization of the dimesitylated-TTM-carbazole isomers. P.G., A.J.G. and A.J.S. performed the three-pulse impulsive vibrational experiment with input from C.S. S.D. synthesized and characterized the radical materials TTM-TPA, TTM-3PCz and TTM-3NCz under the supervision of F.L. P.G., A.R. and R.H.F. co-wrote the manuscript with input from all other authors. R.H.F. and A.R. supervised the work.

**Competing interests** The authors declare no competing interests.

## Additional information
**Correspondence and requests for materials** should be addressed to Akshay Rao.

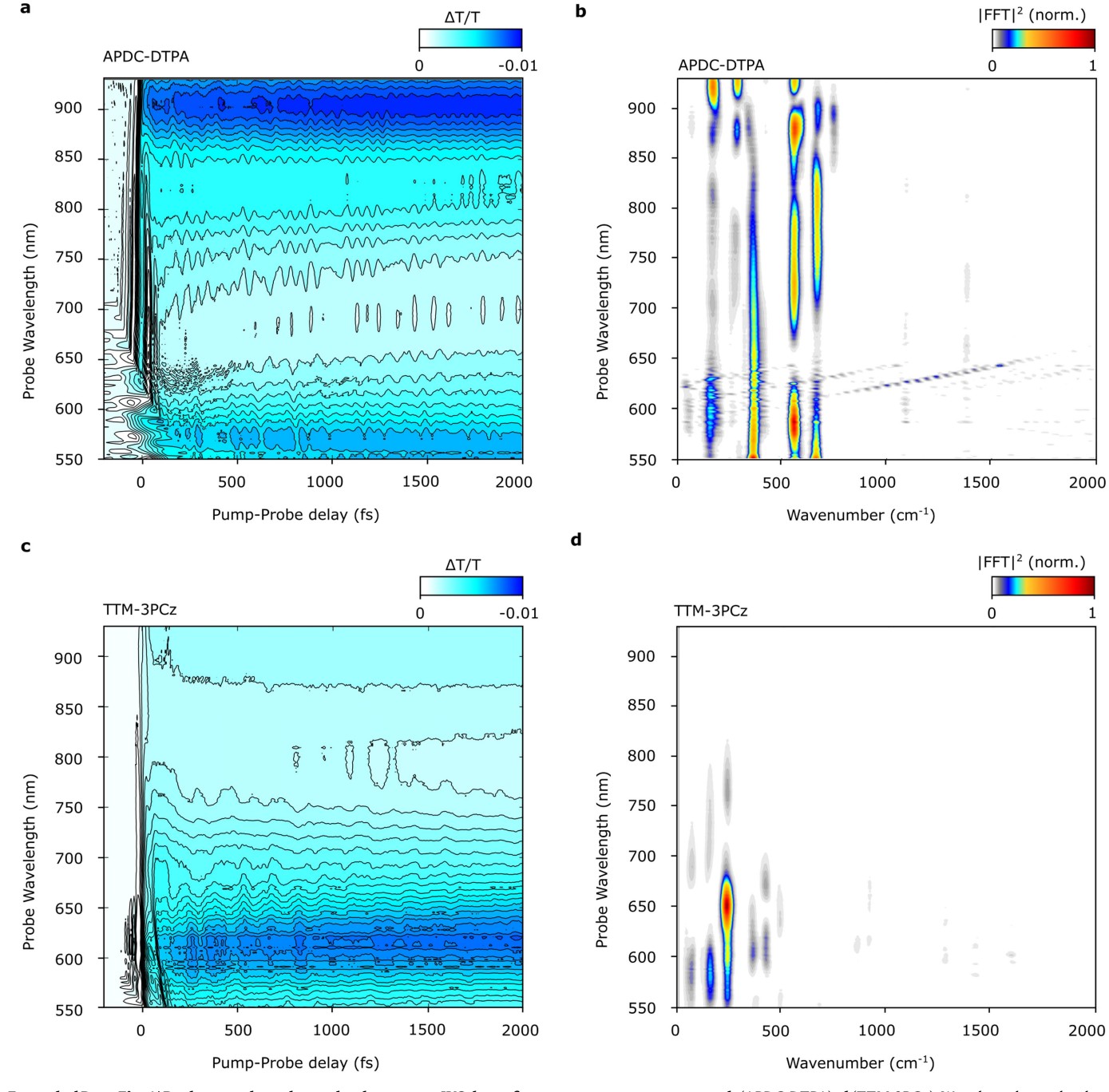

**Extended Data Fig. 1 | Probe-wavelength-resolved resonant-IVS data of APDC-DTPA and TTM-3PCz. a** (APDC-DTPA), **c** (TTM-3PCz), Differential transmission map following excitation with a 10-fs pulse centred at 575 nm, at room temperature. **b** (APDC-DTPA), **d** (TTM-3PCz), Wavelength-resolved impulsive Raman map of following impulsive photo-excitation into lowest excited CT state.

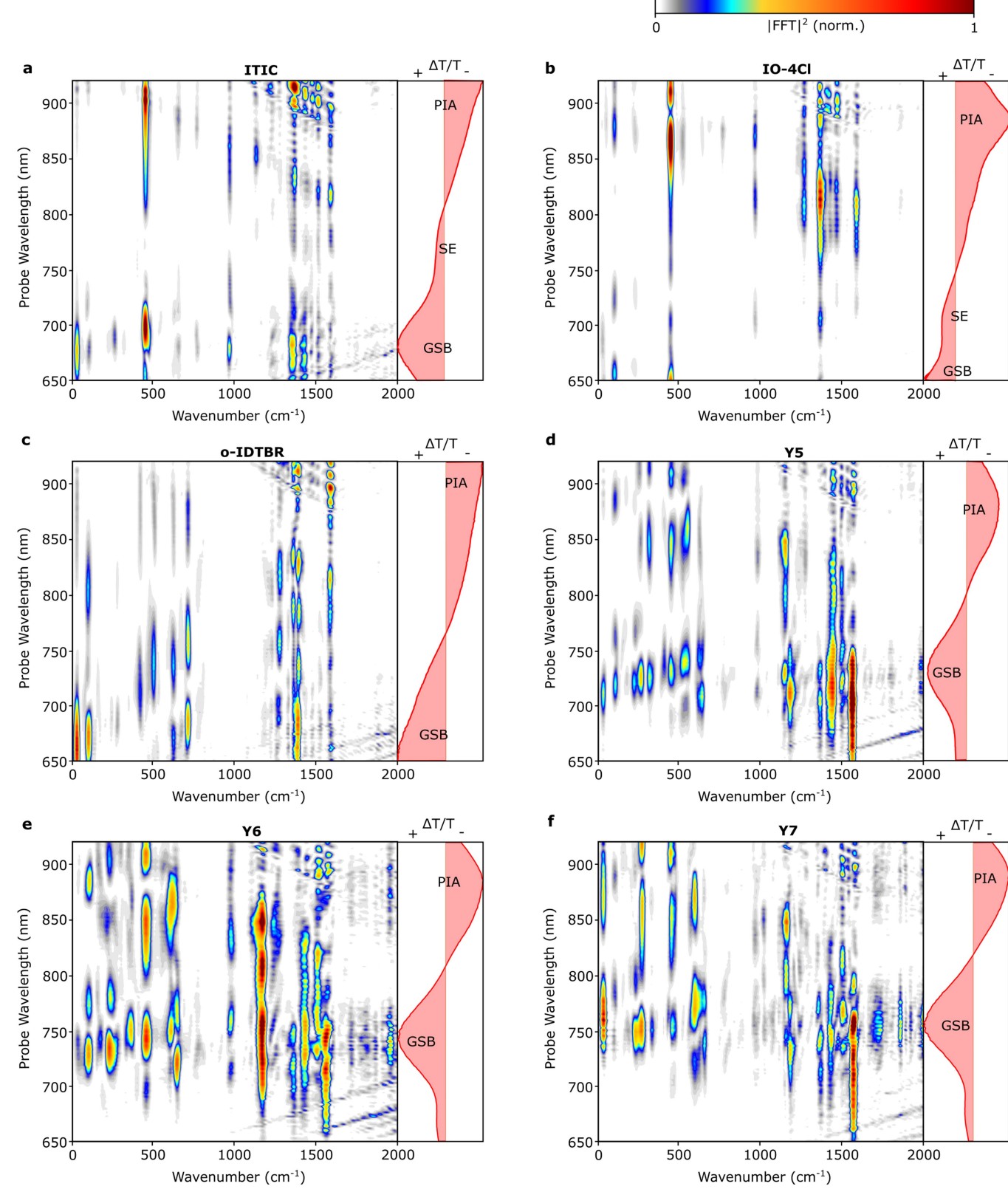

**Extended Data Fig. 2 | Probe-wavelength-resolved resonant-IVS studies on non-fullerene acceptor (NFA) molecules in chloroform solution.** Wavelength-resolved impulsive Raman map of **a** ITIC, **b** IO-4Cl, **c** o-IDTBR, **d** Y5, **e** Y6, **f** Y7. Transient absorption spectra at 1-2 picosecond are plotted in the right inset of every panel. We note that while the NFAs have a donor-acceptor structural motif, due to coplanar geometry and strong electronic conjugation through the fused rings, the HOMO and LUMO strongly overlap in space so that the dipole oscillator strength of the lowest energy transition in these materials is very high, as required for their use in photovoltaics (see the frontier MOs in Supplementary Information section 10). We also note that in many of these molecules the exciton is delocalised across a large spatial extent (greater than for the APDC-DTPA and TTM-3PCz molecules), but that this overall delocalisation does not suppress coupling to high-frequency modes significantly (Supplementary Information section 21) nor reduced resultant non-radiative recombination rate (Fig. 3c).

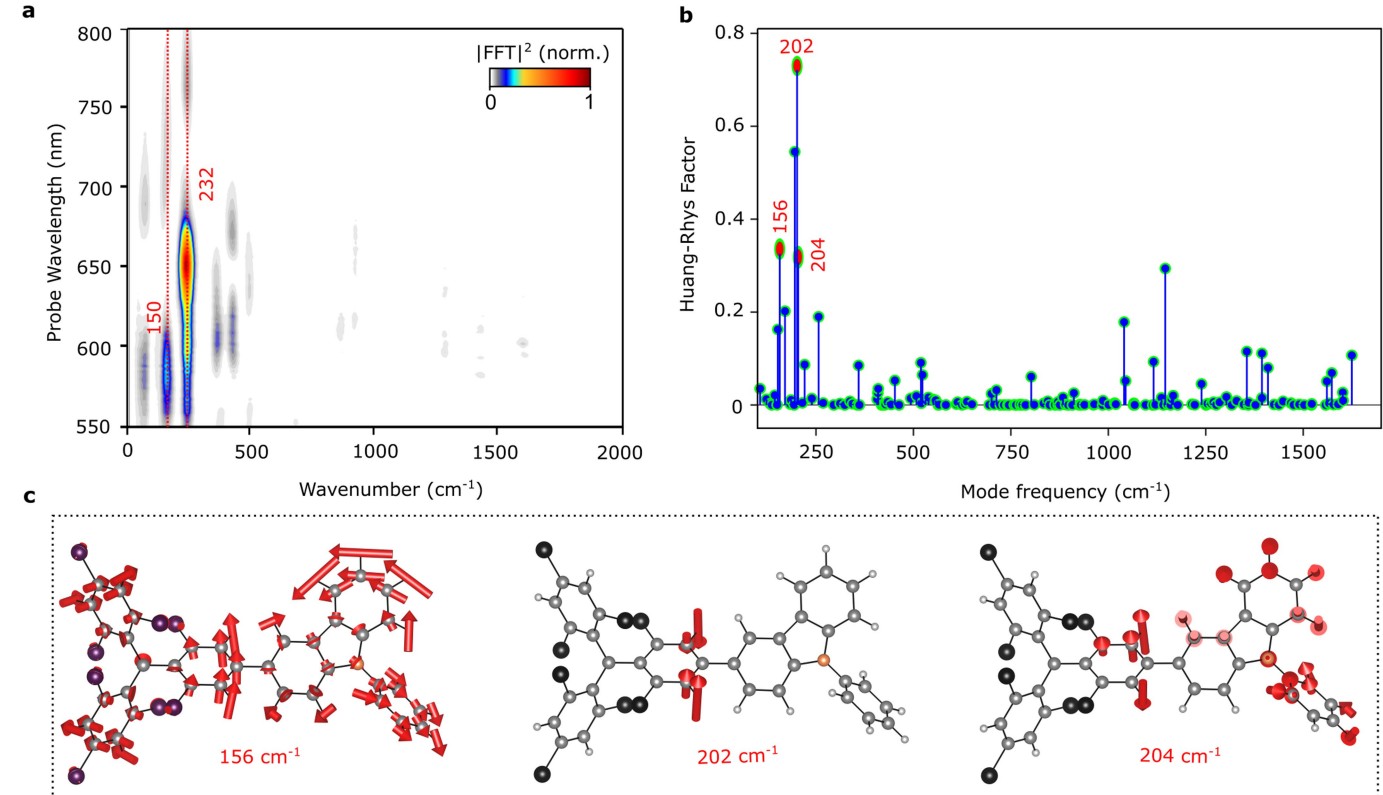

**Extended Data Fig. 3 | Analysis of the exciton-vibration coupling constant of TTM-3PCz and assignment of strongly coupled mode to the excited state. a**, Excited-state Raman map of TTM-3PCz in CHCl$_3$, **b** Calculated Hung-Rhys factor associated with the lowest energy transition (D$_0$ → D$_1$) of TTM-3PCz, **c** vector displacement diagram of the normal modes with calculated frequency 156, 202, 204 cm$^{-1}$ which correspond to experimentally obtained 150, 232 cm$^{-1}$ mode. The description of this figure is discussed in further detail in Supplementary Information (Section 9).

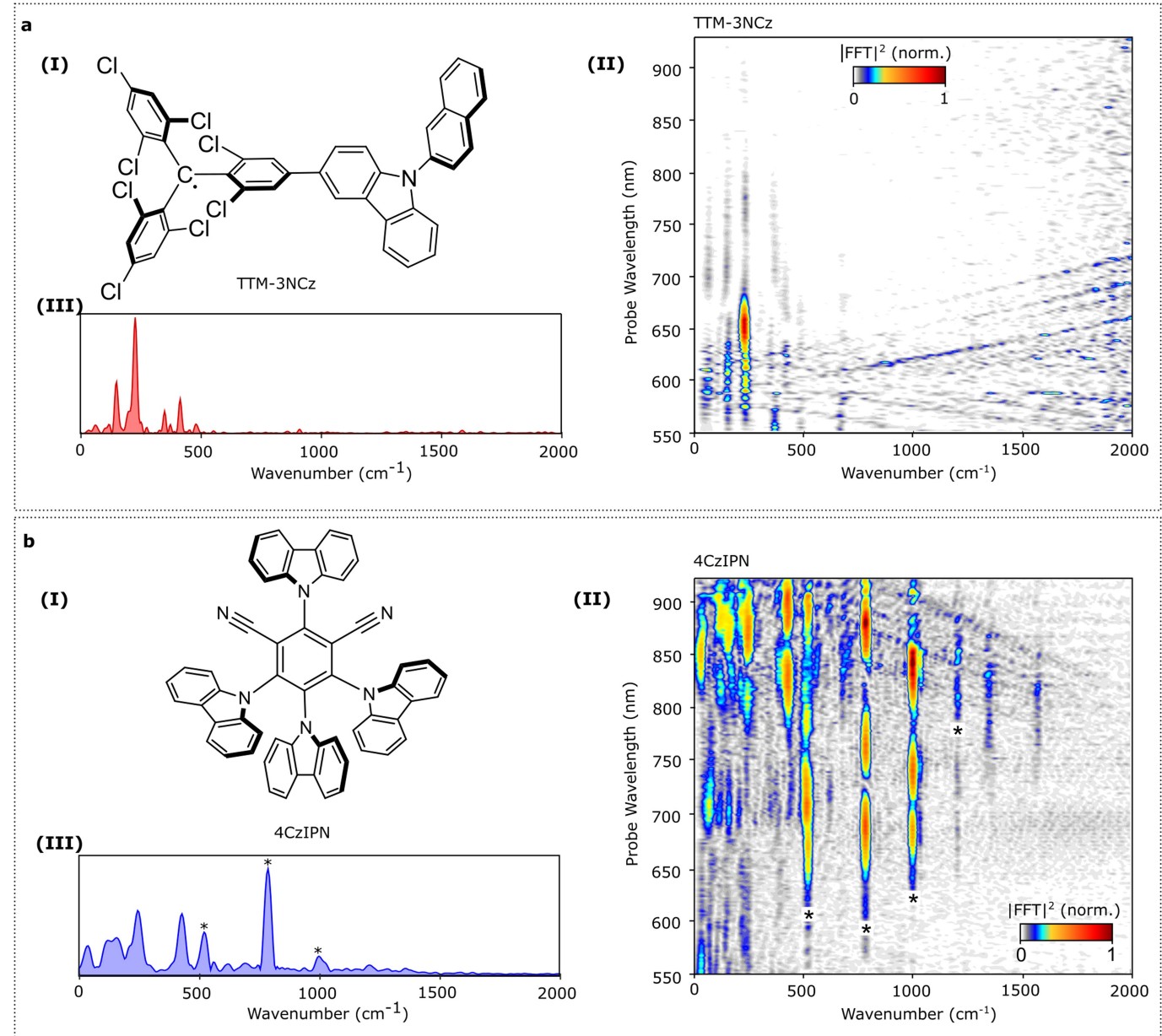

**Extended Data Fig. 4 | Probe-wavelength-resolved IVS studies on a, TTM-3NCz (CHCl₃) and b, 4CzIPN (toluene).** (I) Chemical structure, (II) probe wavelength-resolved impulsive Raman map, (III) excited-state Raman spectra shown for representative probe wavelength (625–635 nm for TTM-3NCz and 870–880 nm for 4CzIPN). Black asterisk (*) corresponds to the solvent modes. The experiment performed on 4CzIPN, is a 3-optical pulse IVS experiment due to the spectral bandwidth limitation of direct impulsive excitation high-energy transitions (>2.4 eV, see methods). 4CzIPN (toluene) is excited to 1CT state with 450 nm (200 fs) pulse followed by inducing vibrational coherence (VC) with 850 nm-centred broadband (8.5 fs) impulsive pulse, 500 fs after initial photo-excitation. The impulsive pulse spectrally overlaps with the excited state absorption of 4CzIPN observed >550 nm. The result shown for 4CzIPN, was recorded in 1 mm pathlength cuvette. This experiment is analysed in further detail in Supplementary information (Section 7).

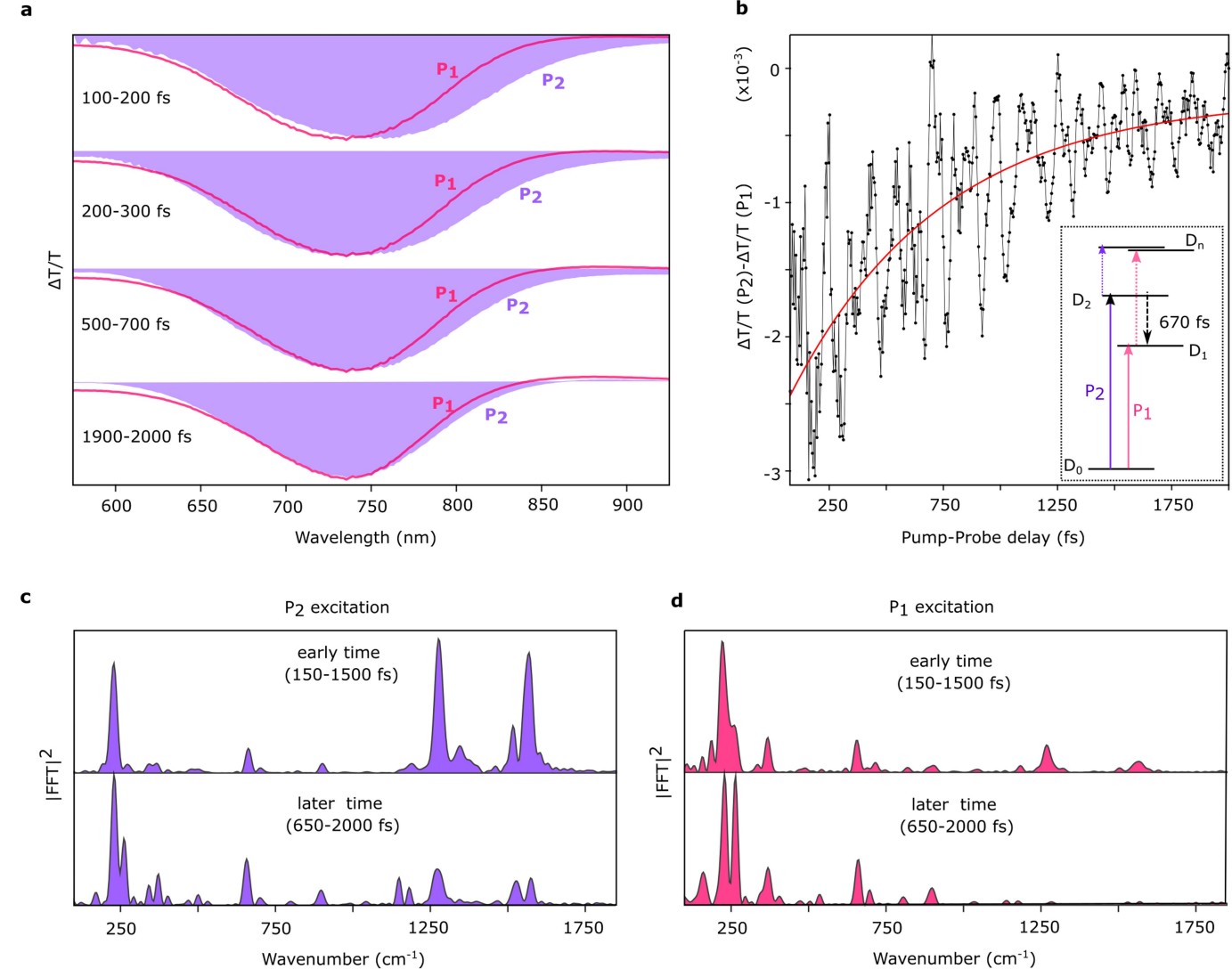

**Extended Data Fig. 5 | Electronic and Vibrational dynamics of internal conversion from $D_2$ to $D_1$ state of TTM-TPA. a)** transient absorption signal in the form of differential transmission ($\Delta T/T$) upon photo-excitation with $P_1$ and $P_2$ pump pulse at different time delays (the purple shaded area: $P_2$ excitation; and magenta line: $P_1$ excitation). **b)** electronic population dynamics of the $D_2$ exciton for a representative wavelength (790–800 nm) which turns out to be $670 \pm 125$ fs. (The method is described in SI section 8). **c, d**, the spectra obtained from the vibrational coherence at early time (150–1500 fs) and later time (650–2000 fs) in band selective excitation experiment with **c)** $P_2$ excitation and **d)** $P_1$ excitation at representative probe wavelength (710–750 nm). It is important to note that a very close resemblance of the later time spectra of $P_2$ excitation and early time spectra of the $P_1$ excitation is consistent with this barrierless and coherent internal conversion process from $D_2$ to $D_1$.

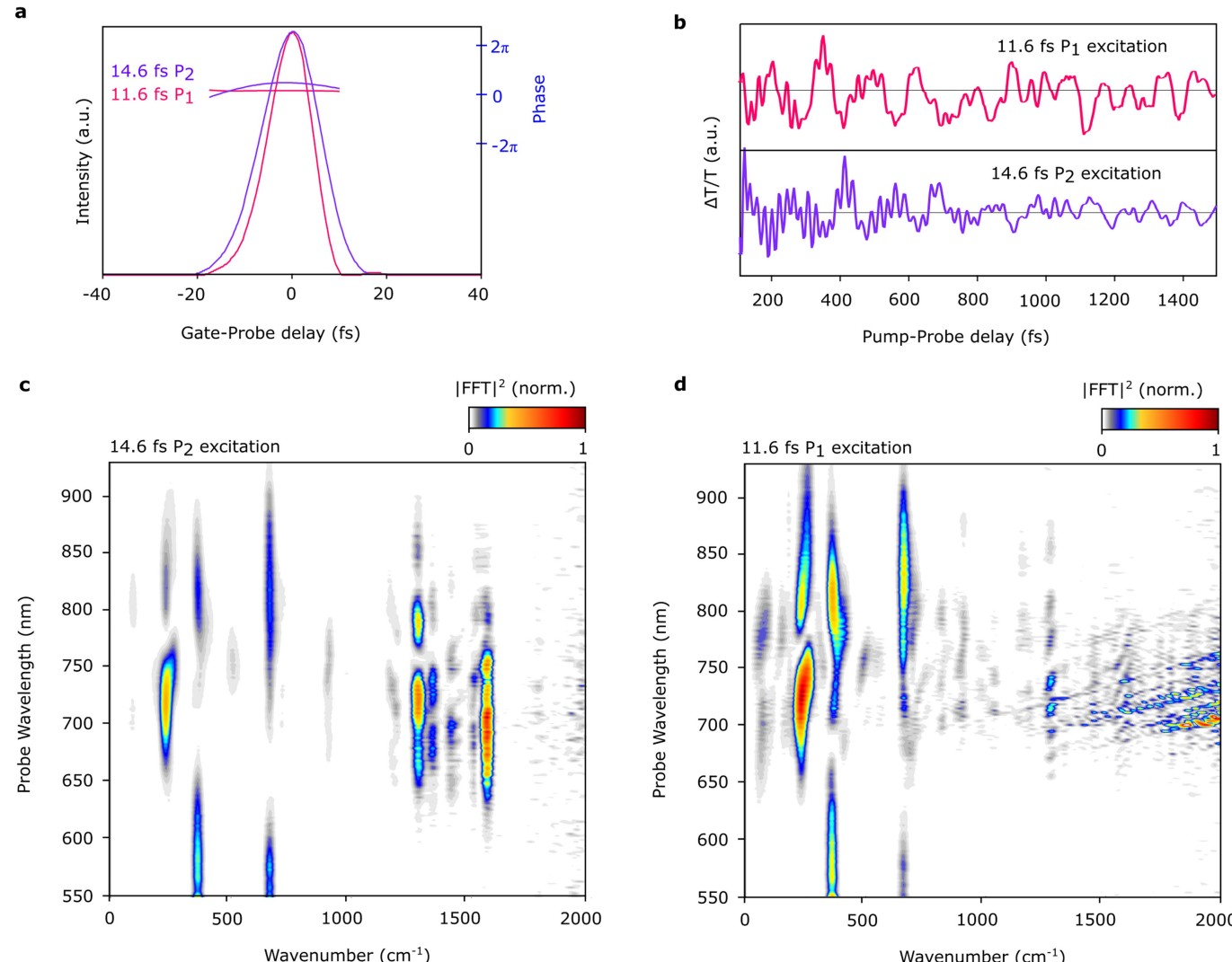

**Extended Data Fig. 6 | Band selective Impulsive vibrational spectroscopy of TTM-TPA. a** Temporal profile of the pulse $P_1$ and $P_2$ used in band selective experiment to excite charge transfer and local exciton-rich excited state respectively. **b**, Vibrational coherence generated after photo-excitation with $P_1$ and $P_2$ pulse for a representative probe window (680–700 nm). **c,d**, Wavelength-resolved impulsive Raman map (time-resolution corrected) of TTM-TPA following impulsive photo-excitation by c) the $P_2$ pulse to the $D_2$ rich state and d) the $P_1$ pulse to the $D_1$ state; The diagonal strips in the d, arise due to the interference scattering with the $P_1$ pump. The fact that, the dominance of the high-frequency oscillations (time-period = 20–30 fs) is highly pronounced in the $D_2$ state even after photo-excited with the slower pump pulse, strongly support the data and calculation presented in the Fig. 2.

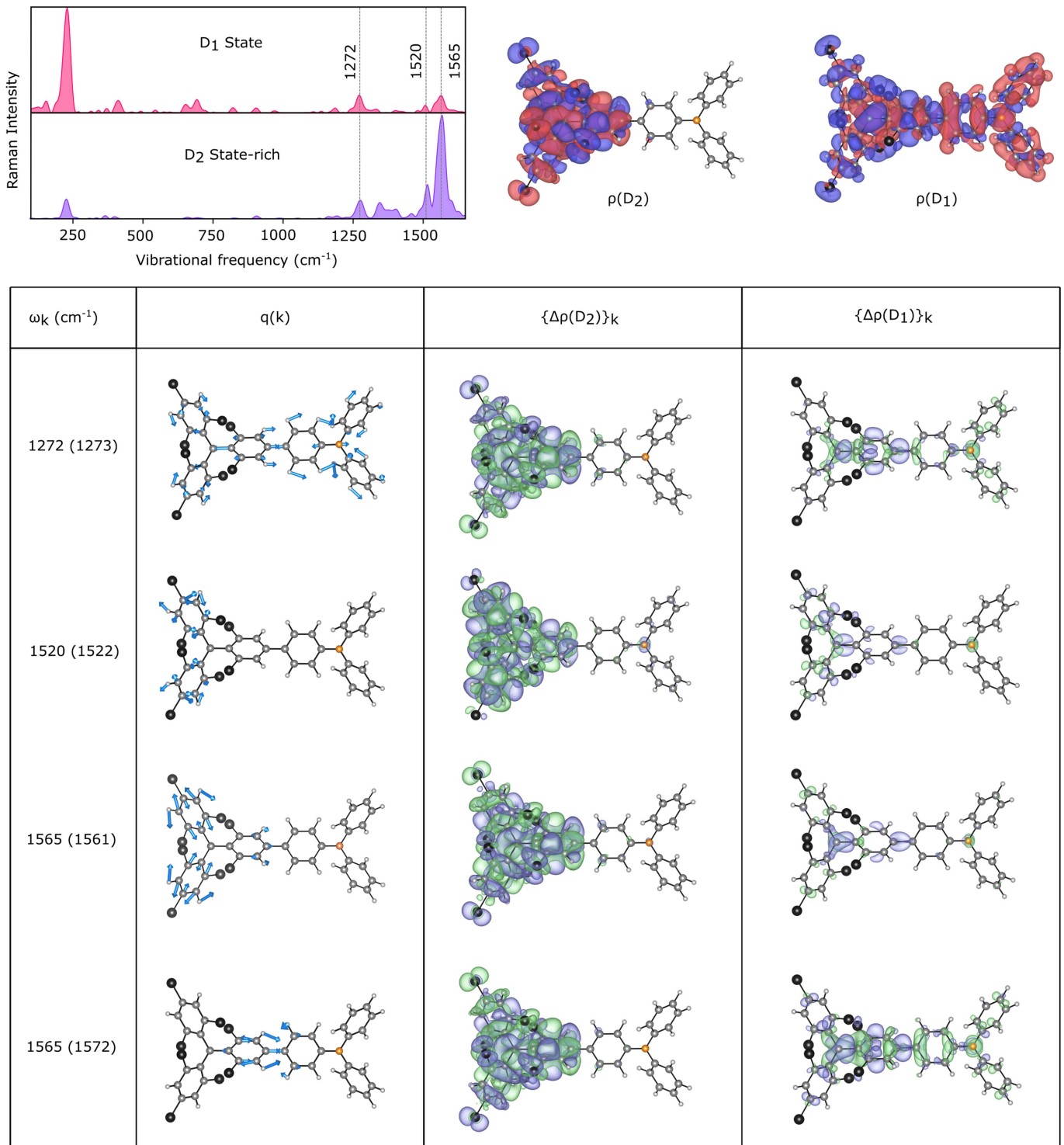

**Extended Data Fig. 7 | Perturbational effect on the exciton wavefunction (transition density) of the $D_2$ and $D_1$ state along the coordinate of all experimentally obtained high-frequency vibrational modes (full mode resolved picture of the data represented in Fig. 3g,h).** on the top left panel, experimentally obtained $D_1$ and $D_2$ state rich Raman spectra are reproduced.

In the table, $\omega_k$(cm$^{-1}$) corresponds to the frequency of the k$^{th}$ vibrational mode. Inside parenthesis, the theoretically obtained frequencies are provided. q(k) corresponds to the vector displacement diagram of the k$^{th}$ mode. $\{\Delta\rho(D_2)\}_k$ and $\{\Delta\rho(D_1)\}_k$ are the differential exciton wavefunction (transition density) upon displacement along the k$^{th}$ mode for $D_2$ and $D_1$ exciton respectively.

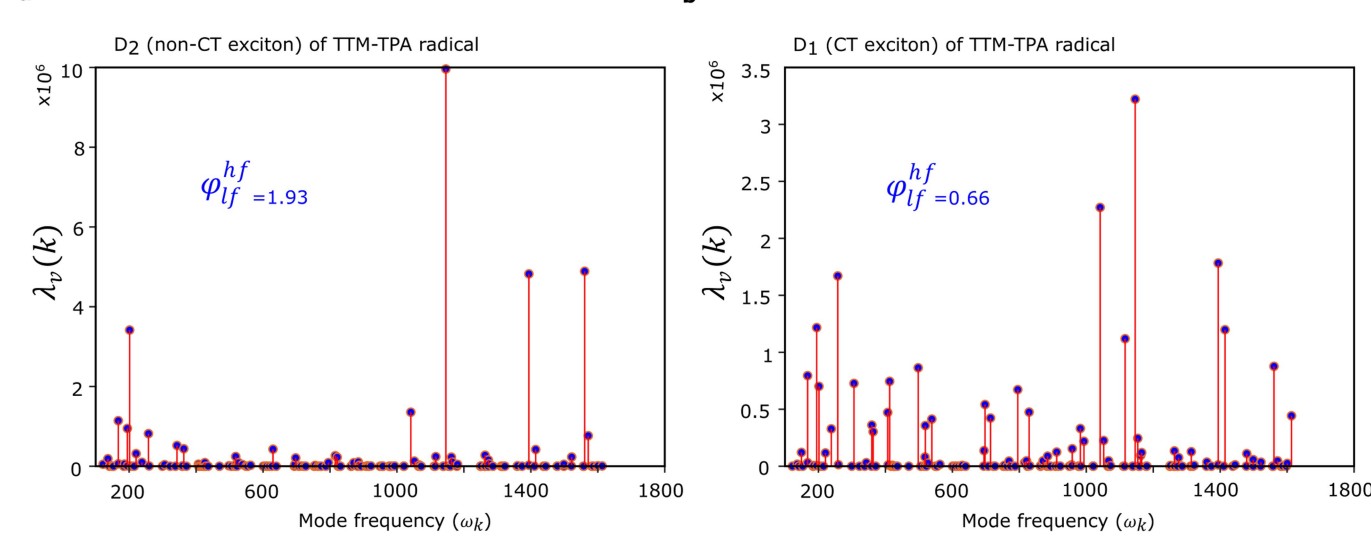

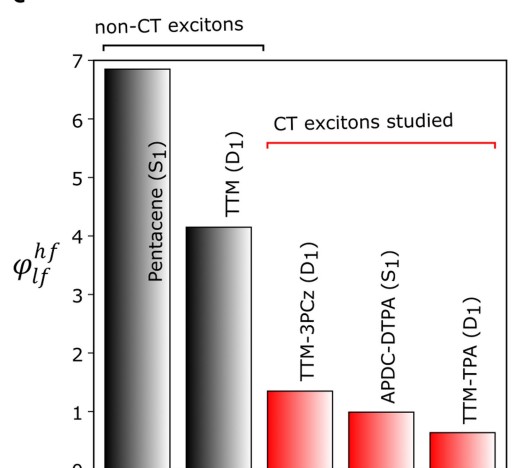

**Extended Data Fig. 8 | Mode-resolved reorganization energy ($\lambda_v$) and contribution of the high-frequency modes to the vibrational reorganization energy. a**, Mode-resolved picture of reorganisation energy for $D_2$ state of TTM-TPA obtained from TDDFT calculation displacing the ground state optimised geometry along the coordinate of each normal modes. **b**, Mode-resolved picture of reorganisation energy for $D_1$ state of TTM-TPA. **c**, $\varphi_{lf}^{hf}$ calculated for lowest excited state of Pentacene ($S_1$, non-CT exciton), TTM radical ($D_1$, non-CT exciton), TTM-3PCz ($D_1$, CT exciton), APDC-DTPA ($S_1$, CT exciton), TTM-TPA ($D_1$, CT exciton). The reduced vibrational coupling to the high frequency regime in TTM-TPA and TTM-3PCz with respect to the TTM presented in c.

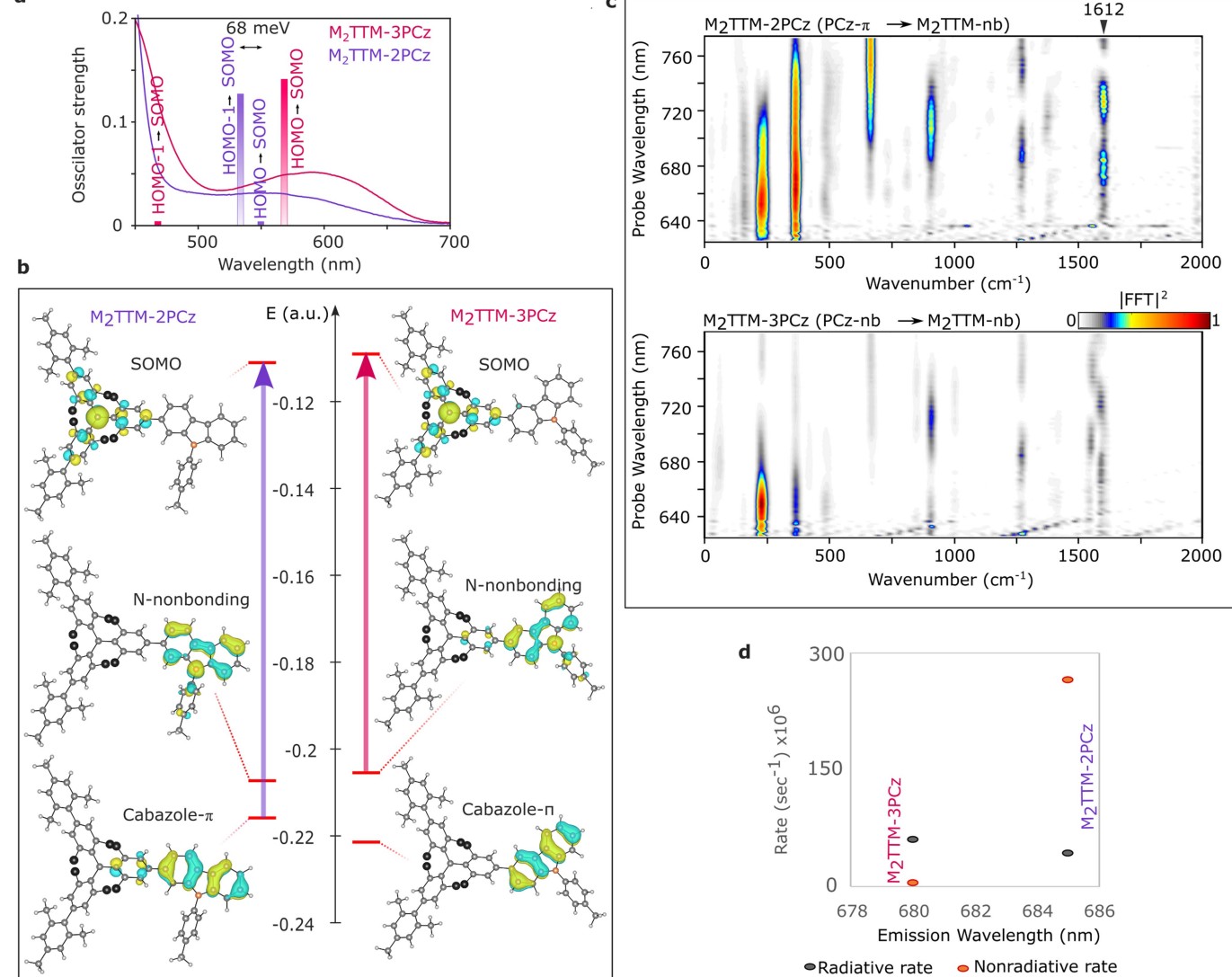

**Extended Data Fig. 9 | Electronic structure and wavelength-resolved impulsive vibrational spectra of M₂TTM-3PCz/2PCz. a**, The transition dipole moment of the two lowest energy electronic transition of the M₂TTM-3PCz and M₂TTM-2PCz obtained from TDDFT with experimental absorption spectra. **b**, Frontier molecular orbitals of the M₂TTM-3PCz and M₂TTM-2PCz and the arrows indicate lowest bright charge-transfer transition. **c**, The wavelength resolved impulsive vibrational maps of M₂TTM-3PCz and M₂TTM-2PCz in CHCl₃. d, radiative and non-radiative rate of the isomers obtained from PLQE and photoluminescence decay rates (see Extended Data Fig. 10 for details). It is important to note that depending on attachment on 2 or 3 positions of the phenyl-carbazole (PCz), the HOMO (N-nonbonding) and HOMO-1 (carbazole-π) molecular orbitals have different extent of leakage into the phenyl ring of the M₂TTM. HOMO-1 (carbazole-π) orbitals is delocalised to the adjacent phenyl

ring of the M₂TTM in M₂TTM-2PCz whereas for M₂TTM-3PCz, HOMO (N-nonbonding) shows the similar phenomenon. N-nonbonding type HOMO for M₂TTM-2PCz and π-type HOMO-1 for M₂TTM-3PCz are extremely localised on the carbazole as the corresponding molecular orbitals have weak orbital coefficient on the linking carbon atom. As the electron accepting level (SOMO) is located on the M₂TTM core, only delocalised occupied orbitals makes charge-transfer electronic transitions with non-negligible transition dipole moment which explains the placement of the arrows in the Extended Data Fig. 9b. Also due to the leakage on the adjacent phenyl ring of M₂TTM, N-nonbonding and the carbazole-π molecular orbitals has higher orbital energy in M₂TTM-3PCz and M₂TTM-2PCz respectively in comparison to alternative isomer (Extended Data Fig. 9b). That explains the smallest energy gap between two lowest energy transitions for M₂TTM-2PCz (Extended Data Fig. 9b).

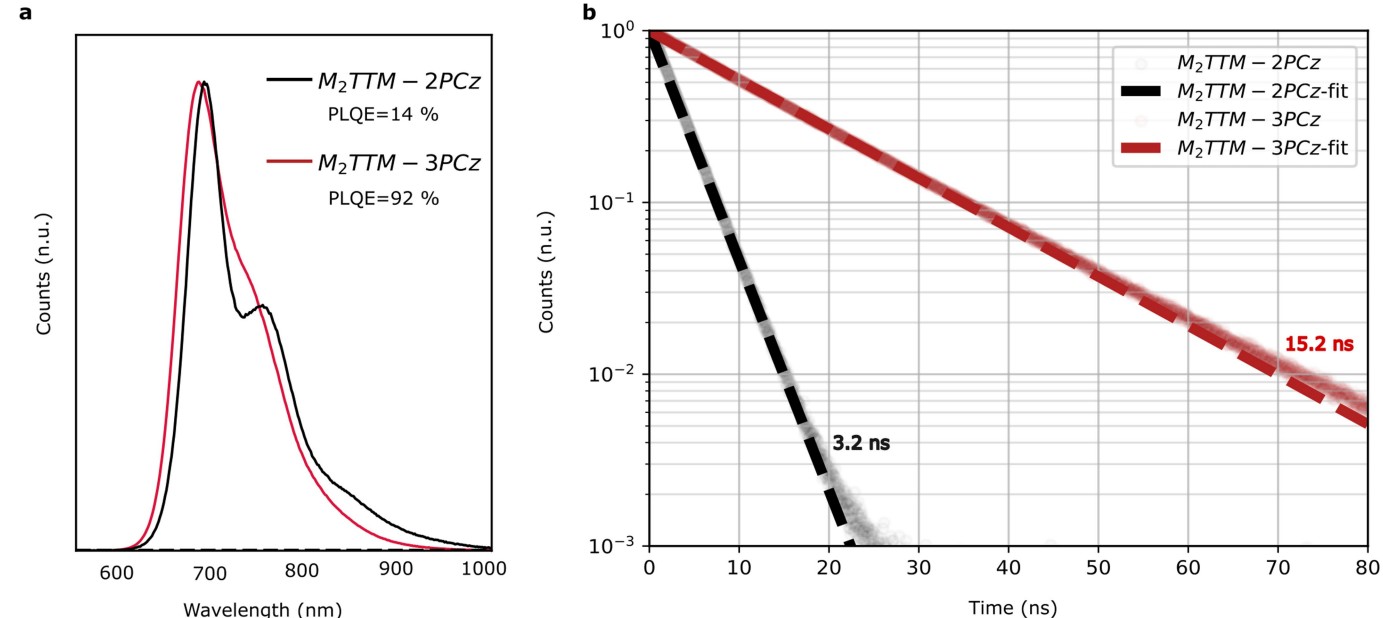

**Extended Data Fig. 10 | PL characterization of M₂TTM-3PCz/2PCz. a**, PL spectra **b**, time-resolved PL decay in solution.

**Extended Data Table 1 | Experimentally obtained non-radiative rates of the studied molecules**

| | System | $\lambda_{em}$ (nm) | $\tau$ (ns) | $\Phi$ (%) | kr ($10^6$ s$^{-1}$) | knr ($10^6$ s$^{-1}$) | $\frac{kr}{knr}$ |
|---|---|---|---|---|---|---|---|
| Deep-red emission | TTM-3PCz | 695 | 22.2 | 65 [a] / 46 [b] | 21 | **24** | 0.85 |
| | TTM-3NCz | 707 | 17.2 | 86 [a] /49 [b] | 28 | **29** | 0.96 |
| | M$_2$TTM-3PCz | 680 | 15.2 | 92 | 61 | **5** | 12.2 |
| | M$_2$TTM-2PCz | 685 | 3.2 | 14 | 43 | **265** | 0.16 |
| | APDC-DTPA | 687 | 11.2 | 63 | 57 | **32** | 1.78 |
| | IO-4Cl | 678 | 1.8 | 36 | 200 | **356** | 0.56 |
| | o-IDTBR | 750 | 1.6 | 19 | 119 | **506** | 0.23 |
| | ITIC | 740 | 0.4 | 2 | 50 | **2450** | 0.02 |
| NIR emission | TTM-TPA | 800 | 8.4 | 24 | 29 | **90** | 0.32 |
| | Y5 | 789 | 0.7 | 5 | 71 | **1357** | 0.05 |
| | Y6 | 797 | 1.1 | 8 | 73 | **836** | 0.08 |

$\lambda_{em}$: PL maxima; $\tau$: PL lifetime; $\Phi$: PLQE; $kr$: radiative rate; $knr$: non-radiative rate. (a: 3wt% in CBP blend, b: Toluene solution). The ratio of kr/knr gives an estimate of how efficient radiative decay pathways are in comparisons to non-radiative decay pathways.

**Extended Data Table 2 | Experimental and calculated (TDDFT) values of the three lowest energy transition of TTM-TPA**

| Experimental Absorption (nm) | Calculated transition energy (nm) | Orbital excitation contribution (%) |
|---|---|---|
| **670** | 718 $(D_0 \rightarrow D_1)$ | 192 beta (HOMO)$\rightarrow$ 193 beta (SOMO): 97% |
| **495** | 463 $(D_0 \rightarrow D_2)$ | 190 beta (HOMO-2) $\rightarrow$ 193 beta (SOMO): 80%<br>189 beta (HOMO-3) $\rightarrow$ 193 beta (SOMO): 10% |
| **433** | 441 | 189 beta (HOMO-3) $\rightarrow$ 193 beta (SOMO): 69%<br>190 beta (HOMO-2) $\rightarrow$ 193 beta (SOMO): 15%<br>192 alpha (HOMO) $\rightarrow$195 alpha (LUMO+1): 4% |

$D_0 \rightarrow D_1$ and $D_0 \rightarrow D_2$ electronic transitions correspond to two lowest energy transitions.