## [Peer Review File · Nature]

Manuscript Title: Decoupling excitons from high-frequency vibrations in organic molecules

Reviewer Comments & Author Rebuttals

Reviewer Reports on the Initial Version:

Referees' comments:

Referee #1 (Remarks to the Author):

The work by Rao et al. shows how to decouple excitons from vibrational modes in organic systems exploiting a combination of ultrafast spectroscopy, quantum calculations and chemical synthesis. This is a really nice manuscript, complete and clear. The authors explain how to control ultrafast energy dissipation by modifying the localization of the molecular vibration of organic molecules in such a way they are not coupled to the electronic transitions. In this way they propose two new design principles and demonstrate them on several molecules. The experiments are performed on relevant molecules for optoelectronics, and also on new systems, making the work quite rich and comprehensive.

The methodology is appropriate and state-of-the-art to provide insight into a general fundamental problem, it represents an original work for the physical chemistry and material science communities and related fields. The results are significant and of immediate interest since they could lead to the development of new materials for applications like NIR OLED or OPVs via molecular synthesis.

The experimental data are clean and very well presented, I enjoyed reading it myself. The level of theory is adequate and needed in order to provide physical insight on the observed mechanism and the chemical synthesis allow them to test the two identified criteria, making the manuscript convincing and with novel results. The interpretation and conclusion are reliable and of broad interest. For these reasons I think this work certainly meets the criteria for being published in a high impact journal.

My only concern for recommending publication in Nature is the following: it is really convincing from their results that CT states are coupled to low frequencies while non-CT excitons to high frequencies. However, I still miss the direct connection with radiative or non-radiative losses that authors suggest in the introduction. The observed vibrational coherences can either participate or not to the internal conversion processes, for instance they can be just spectator modes. A convincing experiment of their findings for me would be the TTM-PA study upon P1 excitation for variable polarity. Increasing the solvent polarity, and thus increasing the CT character, are we going to see higher amplitudes of the low frequencies accompanied by decaying high frequency FT amplitudes and the experimental non radiative rates K_{nr} decreasing? Or, maybe an easier way would be to report the k_{nr} rate and the FT amplitude for different polarity solvents. I think this could be a proof of the concept here, to close the loop which is nicely anticipated in the introduction.

Moreover I have a few more comments/questions that should be addressed before considering publication, here listed.

1. When introducing the energy gap law, I think the term ω in equation 1 holds for the molecular vibrations promoting the transfer. How do the authors assign all the observed vibrations in this category

(see <http://dx.doi.org/10.1063/1.1733413>)? What if they are just spectator modes not participating in the process (for example see ref. 9)? This point should be better explained, since it is a crucial and unsolved aspect both from theoretical and experimental perspectives to explain the old theory of the energy gap law.

2. In reference to the paragraph at lines 149-153: based on their findings, how do they explain the high emission quantum yield in rhodamine? This is coupled to high frequencies, and it possesses higher yield than the CT excitons, which seems to conflict with their statements.

3. The authors cite ref 5. but also in <https://doi.org/10.1038/s41566-022-01079-8> the same authors revisit the theory of the energy gap law to quantify the contribution of each coupled vibrational mode to non-radiative transitions. I think this is also a relevant paper that should be included in the references, since it includes a nice discussion about the role of the active vibrational modes.

Referee #2 (Remarks to the Author):

In organic semiconductors and, more generally, pi-conjugated chromophores, the efficiency of luminescence is intrinsically limited by coupling of the excited state to the vibrations of the molecular backbone. This coupling results in the dissipation of the excited state energy as heat, instead of the desired luminescence. This non-radiative decay channel is exponentially more effective when the excitation couples to high-frequency modes, and when the transition energy is low. In this context, the coupling of an excitation in a pi-conjugated chromophores to the carbon-carbon stretching modes has so far been perceived as an unavoidable and ubiquitous, and hence as an intrinsic limitation to the luminescence efficiency particularly in the red and near infrared spectral range. The authors here demonstrate convincingly a way to overcome this seemingly intrinsic limit. Using a combined optical and theoretical study they show that it is possible to avoid or significantly reduce the coupling of the excitation to the high-frequency carbon-carbon stretching vibrations. The "trick" is to (i) use a donor-acceptor type transition that separates the initial and final state wavefunction, (ii) in a compound that twists after excitation to ensure good wavefunction separation and (iii) preferably involve non-bonding orbitals to further increase wavefunction localization and hence separation. If this is followed, then high-frequency vibrations of the molecular skeleton are not affected by the electronic transition and hence do not couple to it.

This is a very, very nice piece of work, and certainly a very original and significant result. Once laid out, it is very obvious and transparent and one wonders why we could not see this earlier. I certainly like this work and I think it is truly important and appropriately placed in a journal like nature.

The significance of the work lies in the fact that it is a very general and fundamental insight of molecular photophysics. This knowledge allows right away for the design of highly efficient luminescent (or absorbing) compounds in the red and near-infrared spectral range which has so far seemed elusive. As correctly state in the manuscript, such compounds are essential to advance fields such as organic solar cells, red light-emitting diodes, fluorescent biomarkers.

The conclusions presented are supported by the experimental and theoretical data. To me, this work seems robust. It is very rare that I recommend a paper to be accepted without changes, yet in this case I

feel this manuscript is technically correct, and the arguments are clear and elegantly presented, and I cannot see a need for modification.

Hence, I recommend acceptance as is.

Referee #3 (Remarks to the Author):

With decrease in excited-state energy for a molecule, rate constant of excited-state non-radiative decay usually exponentially increases, and PLQY thus significantly decreases, which can be described by so-called energy-gap law. To develop high-efficiency red/NIR photon-absorbing and -emitting organic molecules for OLED and OPV applications, the key is to first provide a fundamental understanding of energy-gap law and then propose molecular-design strategy for violation of energy-gap law. In such context, Rao and co-workers carried out the joint experimental and theoretical investigation to focus on the above fundamental problems, which is important to the community of organic optoelectronics. On the one hand, they calculated Huang-Rhys factors to understand exciton-vibration couplings for different molecular systems; on the other hand, they used broadband Impulsive Vibrational Spectroscopy to probe vibrational coupling in excited state.

I have some comments/questions, as followed:

- 1) In excited state with charge-transfer (CT) excitation, exciton-vibration coupling related to high-frequency vibrations would be reduced. As stated by the authors, it has been reported by previous theoretical calculations. For a CT exciton, its energy can be simply written by $E_{CT} = I_{PDonor} - E_{A} + E_{Coulomb}$. To provide a deep understanding of reduced exciton-vibration coupling for CT exciton, the authors can start from the above equation to rationalize it.
- 2) In non-fullerene acceptor molecules for OPV (such as ITIC, o-IDTBR, Y6, etc), the HOMO and LUMO orbitals are delocalized on almost the whole pi-conjugation backbones. The first singlet excited states for these molecules show weak CT-excitation character. From Fig. 3c, Y6 and Y5 with emission wavelength of ~ 800 nm show similar non-radiative decay rates compared to o-IDTBR with 740 nm emission and IO-4Cl with 680 nm emission. This experimental results cannot be rationalized by the current arguments of the authors. Why? It will impact generality of their conclusions.
- 3) Regarding the point, i.e., reduced exciton-vibration coupling for CT exciton, the authors have examined several different kinds of organic molecules, which is sufficient to exemplify it. However, regarding the point "Non-bonding character of electron and hole levels", the authors just examined two radical systems carefully. So, I suggest the authors examined other different molecular systems (e.g., non-fullerene acceptors for OPV, TADF molecules,, etc) to verify its generality in other systems.
- 4) In Supplementary Figure 16, HOMO and LUMO wavefunctions in triphenylamine both show non-bonding character. Why in LUMO electron-vibration coupling related to high-frequency vibrations cannot be suppressed compared to in HOMO?
- 5) The precision for visualization of molecular orbitals and excited states should be given, since its precision will impact judgement of non-bonding character. In Section 14 in SI, the precision for

visualization of molecular orbitals is not good. The authors need to unify the precision for their visualization.

Author Rebuttals to Initial Comments:

Response to Reviewers (Manuscript: 2022-09-14984A-Z)

We thank all the reviewers for their thoughtful comments, queries, positive comments and constructive critiques. We have tried our very best to carefully address all the questions and suggestions raised by the reviewers.

For the ease of readability:

- *The original reviewers' comments are in italics.*
- our responses are written in blue text
- changes/entries to the manuscript/SI are highlighted with purple text

References are listed at the bottom of the document.

Referee #1 (Remarks to the Author):

The work by Rao et al. shows how to decouple excitons from vibrational modes in organic systems exploiting a combination of ultrafast spectroscopy, quantum calculations and chemical synthesis. This is a really nice manuscript, complete and clear. The authors explain how to control ultrafast energy dissipation by modifying the localization of the molecular vibration of organic molecules in such a way they are not coupled to the electronic transitions. In this way they propose two new design principles and demonstrate them on several molecules. The experiments are performed on relevant molecules for optoelectronics, and also on new systems, making the work quite rich and comprehensive. The methodology is appropriate and state-of-the-art to provide insight into a general fundamental problem, it represents an original work for the physical chemistry and material science communities and related fields. The results are significant and of immediate interest since they could lead to the development of new materials for applications like NIR OLED or OPVs via molecular synthesis. The experimental data are clean and very well presented, I enjoyed reading it myself. The level of theory is adequate and needed in order to provide physical insight on the observed mechanism and the chemical synthesis allow them to test the two identified criteria, making the manuscript convincing and with novel results. The interpretation and conclusion are reliable and of broad interest. For these reasons I think this work certainly meets the criteria for being published in a high impact journal. My only concern for recommending publication in Nature is the following: it is really convincing from their results that CT states are coupled to low frequencies while non-CT excitons to high frequencies. However, I still miss the direct connection with radiative or non-radiative losses that authors suggest in the introduction. The observed vibrational coherences can either participate or not to the internal conversion processes, for instance they can be just spectator modes.

Response: We thank the reviewers for their thoughtful review and comments.

We thank the reviewer for suggesting we provide a more clarified picture of the connectivity between observed vibrational coherence and the non-radiative loss. To address this, we have expanded our analysis to encompass a more careful exploration of the connection between non-radiative transitions from excited (S_1) to ground (S_0) states, induced by vibrational modes, and the vibrational coherence produced by impulsive excitation. This connection is strongly influenced by the dimensionless displacement (Δ) between the minima of the electronic surfaces, particularly in cases of weak electronic coupling limit. We have introduced a new section within the supplementary information and refer to it in the main at relevant places. We believe this new section links the observed electron-vibration coupling through vibrational coherence to the non-radiative decay.

“Supplementary Section 18: mode-resolved Non-radiative loss probed by vibrational coherence:

In the current section we have discussed how the vibrational modes with near zero reorganization energy which behave as spectator modes in the nuclear coordinate of non-radiative transition, cannot be efficiently impulsively generated in the photo-excited wavepacket.

18.1. Non-radiative loss: a mode-resolved picture

Non-radiative transitions from the electronic excited (S_1) to the ground (S_0) state promoted by the vibrational normal modes can be described semi-classically by the energy-gap law^{1,2}. We have applied this approach which is also similar to that used to understand the non-radiative recombination in the organic photovoltaic systems by K. Vandewal^{3,4} and coworkers and J. Nelson and coworkers⁵ where they adopted a model introduced by Jortner⁶ considering the solvents/surrounding modes can be thermally activated while the molecular modes are frozen. Using this concept, the non-radiative rate can be described under the approximation of Fermi’s golden rule and Born-Oppenheimer approximation.

$$k_{nr} \propto V^2 FC$$

V is the electronic coupling between the ground and excited electronic state and the FC is the Frank-Condon wavefunction overlap weighted density of states.

$$FC = (4\pi\lambda_S k_B T)^{-0.5} \sum_{n=0}^{\infty} e^{-S} S^n (n!)^{-1} \exp\left\{-\frac{(g-n\hbar\bar{V}_v-\lambda_S)^2}{4\lambda_S k_B T}\right\}$$

Where $e^{-S} S^n (n!)^{-1}$ describes the wavefunction overlap between 0th vibration state of the excited electronic state and the nth vibrational state of the ground electronic state. S is the Huang-Rhys factor.

The mode-resolved rate equation for the non-radiative transitions derived by Jortner and Englman in the weak electronic coupling limit where no curve crossing happens between the electronic surfaces can be represented as:

$$k_{nr} = \frac{C^2 (2\pi)^{\frac{1}{2}}}{\hbar(\hbar\omega_i \Delta E)^{\frac{1}{2}}} \exp\left[-\frac{\Delta E}{\hbar\omega_i} \left\{\ln\left(\frac{\Delta E}{\sum_i \lambda_i}\right) - 1\right\}\right]$$

Where ω_i is frequency of the 'i'-th mode with reorganization energy λ_i . Chou and coworkers^{7,8} have also used this formalism to understand and reduce non-radiative loss in metal-organic emitters. This equation is frequently referred to the Energy Gap Law.

Here, we categorize all the high and low frequency normal modes into two distinct categories:

1. Spectator modes for non-radiative decay process: These modes exhibit a reorganization energy (λ_i) that is approximately equal to zero. Additionally, the normal coordinates of these modes are orthogonal to the nuclear coordinates of the nonradiative transition. Displacement along these modes has minimal impact on the Frank-Condon overlap, which represents the wavefunction overlap between the vibrational states of the ground and excited states. Consequently, these modes do not significantly contribute to the non-radiative transition.
2. Driving modes for non-radiative decay process: In contrast, driving modes are characterized by non-zero reorganization energies (λ_i) for the nonradiative transition. Along the coordinates of these modes, there is a strong Frank-Condon overlap, signifying their significant involvement in the non-radiative transition process.

In Supplementary Fig. 26 a, we present an analytical model for the evolution of the FC overlap along a vibrational coordinate with the frequency ω , to understand the non-radiative loss behaviour. This analysis follows ref[†] where Benduhn *et al.* performed such an analytical model for the parameter ΔE . We are assuming three different scenarios:

1. non-displaced with low energy-gap (red)
2. non-displaced with high-energy-gap (blue)
3. displaced with high-energy-gap (green).

This visualization illustrates an analytical comprehension of the relationship between non-radiative losses and the Frank-Condon overlap concerning two key parameters: 1. energy-gap (ΔE), 2. dimensionless displacement along a mode (Δ).

Supplementary Fig. 26: **a**, Potential energies of the ground state, $|gs\rangle$ and excited state, $|es\rangle$ as a function of the dimensionless displacement (Δ) along the reaction coordinate a vibrational mode with frequency ω . The $|es\rangle$ surface drawn in red has zero displacement and blue corresponds to the displaced potential surface with respect to the $|gs\rangle$. “m” represents the vibrational states in the $|es\rangle$ manifold and “n” represents the vibrational states in the $|gs\rangle$ manifold. The vibrational wavefunctions (ψ) plotted with the Hermite polynomials. **b**, $\int_{-x}^x \langle \psi_n^* | \psi_0 \rangle^2 dx$ (blue curve) and $\int_{-x}^x \langle \psi_n^* | \psi_0 \rangle dx$ (orange curve) plotted against the vibrational quantum number (n) of the ground state for the non-radiative transition from the $i=0$ ($|es\rangle$) as representative of Frank-Condon factor.; **c**, $\langle \psi_n^* | \psi_0 \rangle^2$ plotted against nuclear coordinate for different vibrational quantum number (n).

The vibrational states in the excited state manifold are denoted by ‘m’ ($m = 0$, in this case) and the vibrational states in the ground electronic state manifold are denoted by ‘n’. As visualised in the Supplementary Fig. 26 b,c, the FC overlap (in the form of $\int_{-x}^x \langle \psi_n^* | \psi_{m=0} \rangle^2 dx$, $\int_{-x}^x \langle \psi_n^* | \psi_{m=0} \rangle dx$, $\langle \psi_n^* | \psi_{m=0} \rangle^2$) increases with lowering the value of the vibrational quantum number, n. This is consistent with the energy-gap law, which predicts the enhancement in the non-radiative loss with lowering the energy-gap, and also with the analytical model of Ref⁴. The effect of the dimensionless displacement (Δ) along any mode is analytically studied with the model under the harmonic approximation. As illustrated in the Supplementary Fig. 27 a-d, we present the wavefunction overlap in the form of $\langle \psi_n^* | \psi_{m=0} \rangle$ and $\langle \psi_n^* | \psi_{m=0} \rangle^2$ for $n=5$ (symmetric state) and $n=8$ (asymmetric state) and we can see a systematic increase in the FC wavefunction overlap with increasing the Δ . $\int_{-x}^x \langle \psi_n^* | \psi_0 \rangle dx$ calculated for $n=2,3,4,10$ (Supplementary Fig. 27 e-h) predicts higher non-radiative rates with increasing the displacement.

Collectively, our analytical model, operating under the harmonic approximation, demonstrates that reducing the displacement between the minima of electronic surfaces, results in a decrease in vibrational Frank-Condon overlap, consequently suppressing non-radiative decay.

Supplementary Fig. 27: **a,b**, $\langle \psi_n^* | \psi_0 \rangle$ and $\langle \psi_n^* | \psi_0 \rangle^2$ for the nonradiative transition between $i=0$ ($|es\rangle$) to $n=5$ ($|gs\rangle$, asymmetric state) in a and b respectively. **c,d**, $\langle \psi_n^* | \psi_0 \rangle$ and $\langle \psi_n^* | \psi_0 \rangle^2$ for the nonradiative transition between $i=0$ ($|es\rangle$) to $n=8$ ($|gs\rangle$, symmetric state) in c and d respectively.; **e,f,g,h**, $\int_{-x}^x \langle \psi_n^* | \psi_0 \rangle^2 dx$ calculated against unitless displacement (Δ) along the mode in the harmonic limit for the non-radiative transition from $i=0$ ($|es\rangle$) to **e**, $n=2$ ($|gs\rangle$); **f**, $n=3$ ($|gs\rangle$); **g**, $n=4$ ($|gs\rangle$); **h**, $n=10$ ($|gs\rangle$);

18.2. Vibrational coherence generated on the excited state manifold with the impulsive vibrational spectroscopy

In the electronic excited state (S_1) manifold, the vibrational ground state is represented as $\psi_{i=0}(x)$ and the first excited vibrational states are $\psi_{i=1}(x)$. After impulsive photoexcitation by a broadband laser pulse from the electronic ground state, multiple vibrational states will be populated, leading to, a newly generated non-stationary states which can be represented as:

$$\psi(r, t) = c_{i=0}(t)e^{-i\omega_0 t}\psi_{i=0}(r) + c_{i=1}(t)e^{-i\omega_1 t}\psi_{i=1}(r) + \dots$$

Where $c_{i=0}$, $c_{i=1}$, corresponds to the contribution of the each vibrational states to the non-eigen state. Hence, for a narrow-band, non-impulsive excitation (no coherent superposition of states) then $c_{i=0} = 1$, $c_{i=1} = 0$. It is noteworthy that the $\psi(r, t)$ has an exponential damping term as well which stands for the dephasing.

The time-dependent molecular polarization can be described as -

$$P(t) = \langle \psi(r, t) | \mu | \psi(r, t) \rangle$$

Where μ is the dipole moment operator.

$$P(t)$$

$$= \langle c_{i=0}(t)e^{-i\omega_0 t}\psi_{i=0}(r) + c_{i=1}(t)e^{-i\omega_1 t}\psi_{i=1}(r) | \mu | c_{i=0}(t)e^{-i\omega_0 t}\psi_{i=0}(r) + c_{i=1}(t)e^{-i\omega_1 t}\psi_{i=1}(r) \rangle$$

$$P(t) = c_{i=0}^* c_{i=1} \mu_{01} e^{-i(\omega_1 - \omega_0)t} + c_{i=1}^* c_{i=0} \mu_{10} e^{-i(\omega_0 - \omega_1)t}$$

$$P(t) = \mu_{01} (c_0^* c_1 e^{-i(\omega_1 - \omega_0)t} + c_1^* c_0 e^{i(\omega_1 - \omega_0)t})$$

The macroscopic polarization can be represented as $P_N(t) = N \cdot P(t)$, where N is the number of molecules⁹. Hence, the vibrational coherence generated by the superposition of two vibrational states can oscillate with $(\omega_1 - \omega_0) = \omega^{les}$. The oscillatory time dependent change in the macroscopic polarization will vanish if the energetic bandwidth of the excitation laser source is lower than ω^{les} . Another key factor for impulsive generation of the wavepacket is during the direct impulsive excitation, the excited surface must have displacement (Δ) with respect to the ground state¹⁰⁻¹².

As a result nuclear wavepacket motion in the multidimensional vibration co-ordinate can be used as a probe for exciton-vibrational coupling. Similar displaced harmonic oscillator model for the band edge exciton transition is also envisioned to quantify the exciton-vibration coupling in inorganic semiconductors by S. Ruhman and coworkers¹³, Tze Chien Sum and coworkers¹⁴. In such impulsive excitation, electron-phonon coupling strengths are represented by a set of parameters [Δ , S , λ]. Δ is a dimensionless displacement of the normal coordinate (as discussed above), S is the Huang-Rhys parameter, and λ is the reorganization energy.

Both S and λ exclusively dependent on the displacement (Δ) as follows:

$$S = \frac{\Delta^2}{2}; \quad \lambda = \hbar\omega S$$

where ω is the frequency of the optical phonon. λ ($\hbar\Delta\omega$) can be calculated from impulsive vibrational spectroscopy data using $A_{OSC} = \left(\frac{dOD}{d\omega}\right) \Delta\omega$, where OD is the optical density of the sample and A_{OSC} is the amplitude of the oscillations. A_{OSC} of any vibrational modes can be obtained by fitting the residuals to a damped sine function. Relative A_{OSC} between different modes for comparison purposes, can be obtained from the relative FFT amplitudes.

Taken together, the spectator modes for a non-radiative transition in the weak coupling limit, which has near zero λ (Δ) cannot efficiently be impulsively generated and are therefore not observed in our experiments. Whereas the driving modes which have larger λ (Δ), Δ and S , can be efficiently generated in the impulsive wavepacket.”

A convincing experiment of their findings for me would be the TTM-PA study upon P1 excitation for variable polarity. Increasing the solvent polarity, and thus increasing the CT character, are we going to see higher amplitudes of the low frequencies accompanied by decaying high frequency FT amplitudes and the experimental non radiative rates K_{nr} decreasing? Or, maybe an easier way would be to report the k_{nr} rate and the FT amplitude for different polarity solvents. I think this could be a proof of the concept here, to close the loop which is nicely anticipated in the introduction.

Response: We thank the reviewer for their suggestion to perform solvent polarity dependent impulsive vibrational spectroscopy of TTM-TPA pumped with the P1 pulse to further strengthen our work.

We agree with the reviewer that due to the increased solvent polarity, the photoexcited state will have a stronger charge-transfer character in the Frank-Condon state. We therefore performed broadband impulsive vibrational spectroscopy on the TTM-TPA with P1 as suggested by the reviewer for the solvents p-Xylene<Toluene<Chloroform (in the increasing order of the solvent polarity¹) and we observed a systematic decrease in the contribution of the high-frequency modes ($>1000\text{ cm}^{-1}$) to the exciton-vibrational coupling with increasing solvent polarity. We have now added a new section in supplementary information to incorporate this dataset and cited that in the relevant parts in the main text.

“Section 19: Impulsive vibration spectroscopy of TTM-TPA in variable solvent polarity

Supplementary Fig. 28 Solvent dependent IVS (P1) on TTM-TPA. a, IVS data of the TTM-TPA in different solvents (solvent polarizability: p-Xylene<Toluene<Chloroform). Integrated Fourier transformed spectra for chloroform, toluene, p-Xylene (Integrated over $\lambda = 530 - 750\text{ nm}$ (red), $\lambda = 650 - 850\text{ nm}$ (black)) **b**, Intensity ratio of $|FFT|^2(1275\text{ cm}^{-1})$ to $|FFT|^2(230\text{ cm}^{-1})$ plotted against the relative polarity of the solvent used (red data). For

¹ Representative relative polarity (relative to water) (P) of the solvents used are P(p-Xylene): 0.074, P(Toluene): 0.099; P(Chloroform): 0.259. Data taken from <http://murov.info/orgsolvsort.htm>

reference dielectric constant of the solvent is also plotted (blue). Modes that are asterisked are off-resonant contribution of the solvent modes.

To further investigate how the degree of charge-transfer plays a role in the radical emitter system, we performed resonant broadband impulsive vibrational spectroscopy on TTM-TPA in 3 solvents with varying polarity (relative polarity order: p-Xylene<Toluene<Chloroform), using P1 to impulsively excite the $D_0 \rightarrow D_1$ transition.

As shown in Supplementary Fig. 28a, we find that high frequency modes (e.g. 1275 cm^{-1}) for the most polar solvent (Chloroform) exhibit minimum intensity relative to the low frequency modes. In the least polar solvent (p-Xylene), the same high-frequency mode shows a higher intensity relative to the low-frequency modes. The trend showcased in Supplementary Figure 28b aligns with a systematic decrease in the ratio of the contribution of the representative high-frequency mode at 1275 cm^{-1} to the contribution of the representative low-frequency mode at 230 cm^{-1} as the solvent's relative polarity and dielectric constant increase. These findings further support that the suppression of exciton coupling to high-frequency, localized C-C stretching modes occurs when the exciton exhibits charge-transfer characteristics involving spatially-separated electron-hole pairs.”

We note that the reviewer requested to correlate the relative FT amplitude with Kn_r . While in principle this would be preferred, several system-specific factors prevent us from extending the polarity dependence to a quantitative comparison with Kn_r .

The main limitation arises from the solvent-polarity dependence of the system's Stokes shift, as shown in the figure from [arXiv:2308.02355](https://arxiv.org/abs/2308.02355), which is also exhibited by similar CT-type radical emitter^{15,16}. As the polarity increases the emission spectra significantly red-shift (λ_{em} (p-Xylene)=786 nm; λ_{em} (Toluene)=800 nm; λ_{em} (ChCl₃)=860 nm), leading to a changes in the system's energy gap, which counteracts the high-frequency trend in more polar solvents shown above.

“PL and absorption spectra of TTM-TPA in different solvents. **a**, Steady-state ultraviolet-visible and **b**, photoluminescence spectra of TTM-TPA (0.1 mg/ml) in various solvents with various polarizability index, f. Photo excitation wavelength is 532 nm. **c**, Lippert–Mataga plot of the Stokes shift ($v_a - v_f$) versus f for TTM-TPA (circles). v_a and v_f denote the absorption and fluorescence energies, respectively.”

This figure(with caption) is the part of a working paper, Cho, H. H., Gorgon, S., Londi, G., Giannini, S., Cho, C., Ghosh, P., Tonnele C., Casanova D., Oliver Y., Li F., Beljonne D., Greenham C. N., Friend H. R.,

& Evans, E. W. (2023). Efficient near-infrared organic light-emitting diodes with emission from spin doublet excitons. arXiv preprint arXiv:2308.02355.

We continued discussing this and other aspects in the supplementary section:

“Within the framework of the 'energy-gap law' rate expression, it is crucial to recognize that the non-radiative rate (K_{nr}) is influenced by both, the emission energy gap (ΔE) and the frequency of strongly coupled vibrational modes.

As illustrated in Supplementary Fig. 29 a, the impact of the energy-gap law on K_{nr} is significantly reduced in radical systems compared to e.g. NFA systems, due to the decoupling of high-frequency modes. However, the energy gap law still applies! Consequently, K_{nr} depends on two contrasting effects: a) a decreasing trend in K_{nr} due to the reduced coupling of high-frequency modes with increasing charge-transfer character and b) an increasing trend in K_{nr} resulting from the lower energy gap due to a higher Stokes shift in more polar solvents.

This complication is further supported by previous work¹⁶ on a similar class of TTM-based radicals, where the donor was pyridoindolyl (PyID) instead of triphenylamine (TPA), where it was found that K_{nr} is largely independent of solvent polarity, as depicted in Supplementary Fig. 29 b. Therefore, the interplay of these two competing effects on non-radiative losses with varying solvent polarity can elucidate the observed trends.

Supplementary Fig. 29: a, Non-radiative rate for NFA-systems with similar structure (IO-4Cl, o-IDTBR) and radical systems (TTM-3PCz, TTM-3NCz and TTM-TPA). b, Non-radiative rate plotted for TTM-Carbazole radicals TTM-pyridoindolyl derivatives: TTM- α PyID, TTM- β PyID and TTM- δ PyID in different solvents with variable polarity. (the data points that are highlighted in red are the solvents which are used for solvent-dependent IVS of TTM-TPA). This dataset are taken from the reference¹⁶.

Another technical aspects of this experiments that is noteworthy are as follows:

1. Despite retaining the same P1 pump spectra for all experiments, variations arise in the absorption cross-section between absorption spectra and pump spectra due to the solvatochromic shift (Supplementary Fig. 30). Hence, the coefficient $c_{i=1}(t)$ in the wavepacket equation, will exhibit sample-to-sample variations in the impulsive population of a high-frequency mode leading to highest impulsive population of the high-frequency mode for the solvent with broadest absorption cross-section.

$$\psi(r, t) = c_{i=0}(t)e^{-i\omega_0 t}\psi_{i=0}(r) + c_{i=1}(t)e^{-i\omega_1 t}\psi_{i=1}(r) \quad (\text{wavepacket equation}).$$

Intriguingly, in Chloroform the overlap between pump and absorption spectra is broadest. Therefore, if we could correct for this effect, the trend depicted in Supplementary Figure 28 would gain even more significance.

2. The differences in the absorption cross-sections in different solvents, leads to minute variation in the pump beam's penetration depth from the front surface of the cuvette. This variation leads to varying pulse dispersion, with the most significant dispersion occurring in less absorbing samples. Consequently, it's crucial to recognize that the effective time-resolution will slightly differ across various samples, particularly in the context of quantifying the absolute intensity of high-frequency modes. To minimise this

effect we performed our solvent dependent IVS experiments in ultrathin (200 micrometre path length cuvette).

Supplementary Fig. 30: The plots depicts the overlapped area between the P1-pump spectra and the absorption spectra of TTM-TPA in different solvents

Moreover I have a few more comments/questions that should be addressed before considering publication, here listed.

1. When introducing the energy gap law, I think the term ω in equation 1 holds for the molecular vibrations promoting the transfer. How do the authors assign all the observed vibrations in this category (see <http://dx.doi.org/10.1063/1.1733413>)? What if they are just spectator modes not participating in the process (for example see ref. 9)? This point should be better explained, since it is a crucial and unsolved aspect both from theoretical and experimental perspectives to explain the old theory of the energy gap law.

Response: The energy gap law as written in eq. (1) of our manuscript relies on several approximations that allow one to write a rate equation based on Fermi's golden rule. Within this picture, the sole determining factor of whether a vibrational mode i promotes non-radiative recombination, is its reorganisation energy λ_i . Vibrational mode with near zero reorganization energy has very little effect on non-radiative loss. Specifically, we have,

$$k_{nr} = \frac{C^2(2\pi)^{\frac{1}{2}}}{\hbar(\hbar\omega_i\Delta E)^{\frac{1}{2}}} \exp\left[-\frac{\Delta E}{\hbar\omega_i}\left\{\ln\left(\frac{\Delta E}{\sum_i \lambda_i}\right)-1\right\}\right]$$

We agree with the referee that more complex effects could potentially have an effect on the relaxation of the excited state to the ground state. Such effects could emerge as an aspect of the vibrational relaxation process, where vibrational modes exchange energy with each other, and are beyond what a rate equation can describe, requiring the explicit accounting of memory

effects beyond the so-called Markov approximation. Accounting for such vibrational relaxation effects can indeed be important for describing the internal conversion between excited states, which has a timescale of the order of a picosecond, similar to the timescale of the excited state dynamics, as shown in ref¹⁷. However, the non-radiative decay from the excited to the ground state occurs on much slower timescales (nanoseconds or slower), which suggests that the vibrational relaxation process is long complete, having reached an equilibrium for the mode populations. As a result, the dominant factor in determining the coupling of the excited to the ground state, is the coupling constants of the electronic and vibrational subsystems, described by the reorganization energy. This is in agreement with the argument put forward in the reference that the referee has provided.

While we agree that it would be interesting to examine the role of vibrational relaxation in the non-radiative recombination process, the accurate methods that commonly allow to go beyond the Markov approximation and include memory effects, such as tensor network methods¹⁷, typically can only access timescales within the picosecond timescale, making the modelling of slower processes very challenging.

Additionally, in the main text we specify that we only consider the driving modes with non-zero reorganization energy in the context of energy-gap law:

$$k_{nr} = \frac{C^2(2\pi)^{\frac{1}{2}}}{\hbar(\hbar\omega\Delta E)^{\frac{1}{2}}} \exp\left[-\frac{\Delta E}{\hbar\omega}\left(\ln\left(\frac{\Delta E}{\sum_i \lambda_i}\right)-1\right)\right] \quad (1)$$

Where, C is the effective electronic coupling matrix element and λ_i corresponds to the reorganization energy associated with **the driving modes that promote¹⁸ non-radiative relaxation.**”

2. In reference to the paragraph at lines 149-153: based on their findings, how do they explain the high emission quantum yield in rhodamine? This is coupled to high frequencies, and it possesses higher yield than the CT excitons, which seems to conflict with their statements.

Response: We appreciate the referee's astute observation regarding the apparent contradiction between the strong exciton-vibration coupling in the high-frequency regime and the high photoluminescence quantum yield exhibited by Rhodamine 6G, a well-known laser dye. We apologize for any lack of clarity in our earlier statement. What we intended to convey is that Rhodamine 6G exhibits emission at 530 nm (corresponding to a band gap energy of 2.34 eV), which is notably higher than the band gap energies associated with the low-band gap NIR chromophores studied in this work. In essence, this elevated band gap of Rhodamine 6G means that its emission should be less susceptible to the influence of high-frequency vibrational modes when compared to the lower-band gap chromophores according to the ‘energy-gap law’ in the context of the multi-phonon emission¹⁹ or less efficient vibrational wavefunction overlap in the high bandgap systems⁴.

We made the following change in the main manuscript to clarify this point:

“the laser dye rhodamine 6G (r6G), which emits brightly **at relatively higher energy gap** (PLQE 94%, 530 nm).”

We also made an addition in the Fig 1 where rr-P3HT, TTM-3NCz and TTM-3PCz are color-coded in red and highlighted in the category of ‘low energy-gap’ systems where Rhodamine 6G is color-coded in blue and highlighted in the category of ‘high energy-gap’ system.

Fig. 1 (revised)

The figure caption is modified as:

“...The name of the molecule is highlighted in red if the corresponding emission maxima are above 650 nm. The maximum reported PLQEs, Φ , are indicated. **It is noteworthy that Rhodamine-6G has high PLQE even after possessing strong coupling to high-frequency modes due to the high-energy gap emission.**”

3. The authors cite ref 5. but also in <https://doi.org/10.1038/s41566-022-01079-8> the same authors revisit the theory of the energy gap law to quantify the contribution of each coupled vibrational mode to non-radiative transitions. I think this is also a relevant paper that should be included in the references, since it includes a nice discussion about the role of the active vibrational modes.

Response: We thank the referee for suggesting the inclusion of the above mentioned paper ref⁷ (*Nature Photonics*, 16(12), 843-850) which gives a well-described picture of the promoting mode in a non-radiative transition. The authors of the referenced work have demonstrated the

dependence of the modulation of the ‘active energy-gap’ on the frequency of the promoting mode. Furthermore, engineering frequency of the mode through isotope substitution leads to reduction in the FC overlap and suppression of the corresponding non-radiative rates. This work is certainly highly relevant to the discussion of the current paper. We have promptly cited this paper in current paper while discussing the mode-resolved energy-gap law.

“.....and λ_i corresponds to the reorganization energy associated with the driving modes that promote⁶ non-radiative relaxation.”

6. Wang, S. F. *et al.* Polyatomic molecules with emission quantum yields >20% enable efficient organic light-emitting diodes in the NIR(II) window. *Nat Photonics* **16**, 843–850 (2022).

Referee#2 (Remarks to the Author):

In organic semiconductors and, more generally, pi-conjugated chromophores, the efficiency of luminescence is intrinsically limited by coupling of the excited state to the vibrations of the molecular backbone. This coupling results in the dissipation of the excited state energy as heat, instead of the desired luminescence. This non-radiative decay channel is exponentially more effective when the excitation couples to high-frequency modes, and when the transition energy is low. In this context, the coupling of an excitation in a pi-conjugated chromophores to the carbon-carbon stretching modes has so far been perceived as an unavoidable and ubiquitous, and hence as an intrinsic limitation to the luminescence efficiency particularly in the red and near infrared spectral range. The authors here demonstrate convincingly a way to overcome this seemingly intrinsic limit. Using a combined optical and theoretical study they show that it is possible to avoid or significantly reduce the coupling of the excitation to the high-frequency carbon-carbon stretching vibrations. The “trick” is to (i) use a donor-acceptor type transition that separates the initial and final state wavefunction, (ii) in a compound that twists after excitation to ensure good wavefunction separation and (iii) preferably involve non-bonding orbitals to further increase wavefunction localization and hence separation. If this is followed, then high-frequency vibrations of the molecular skeleton are not affected by the electronic transition and hence do not couple to it.

This is a very, very nice piece of work, and certainly a very original and significant result. Once laid out, it is very obvious and transparent and one wonders why we could not see this earlier. I certainly like this work and I think it is truly important and appropriately placed in a journal like nature.

The significance of the work lies in the fact that it is a very general and fundamental insight of molecular photophysics. This knowledge allows right away for the design of highly efficient luminescent (or absorbing) compounds in the red and near-infrared spectral range which has so far seemed elusive. As correctly state in the manuscript, such compounds are essential to advance fields such as organic solar cells, red light-emitting diodes, fluorescent biomarkers. The conclusions presented are supported by the experimental and theoretical data. To me, this work seems robust. It is very rare that I recommend a paper to be accepted without changes, yet in this case I feel this manuscript is technically correct, and the arguments are clear and elegantly presented, and I cannot see a need for modification. Hence, I recommend acceptance as is.

Response: We thank the reviewer for their very positive and thoughtful review of our paper, emphasising its significance and quality for a large number of highly active fields.

Referee #3 (Remarks to the Author):

With decrease in excited-state energy for a molecule, rate constant of excited-state non-radiative decay usually exponentially increases, and PLQY thus significantly decreases, which can be described by so-called energy-gap law. To develop high-efficiency red/NIR photon-absorbing and -emitting organic molecules for OLED and OPV applications, the key is to first provide a fundamental understanding of energy-gap law and then propose molecular-design strategy for violation of energy-gap law. In such context, Rao and co-workers carried out the joint experimental and theoretical investigation to focus on the above fundamental problems, which is important to the community of organic optoelectronics. On the one hand, they calculated Huang-Rhys factors to understand exciton-vibration couplings for different molecular systems; on the other hand, they used broadband Impulsive Vibrational Spectroscopy to probe vibrational coupling in excited state.

I have some comments/questions, as followed:

1) In excited state with charge-transfer (CT) excitation, exciton-vibration coupling related to high-frequency vibrations would be reduced. As stated by the authors, it has been reported by previous theoretical calculations. For a CT exciton, its energy can be simply written by $E_{CT} = IP_{Donor} - EA_{Acceptor} + E_{Coulomb}$. To provide a deep understanding of reduced exciton-vibration coupling for CT exciton, the authors can start from the above equation to rationalize it.

Response: We thank the reviewers for their thoughtful review and comments.

We agree with the reviewer's insightful observation that there exist valuable model expressions for the energy of excited states, particularly those involving charge transfer (CT) excitons, as exemplified by the reference provided by the reviewer. These expressions can significantly contribute to the elucidation of observed trends, wherein the degree of charge transfer character plays a pivotal role in determining the strength of coupling to high-frequency vibrations. We have now expanded upon this critical aspect in the supplementary information of our manuscript. Additionally, we have included references to this analysis within the main body of our paper. In summary, our supplementary information now offers a comprehensive elaboration of the following key arguments:

Section 20: Extended discussion on dependence of exciton-vibration coupling on degree of charge transfer character

Following ref²⁰, the energy of a local exciton may be written as:

$$E_{LE} = \Delta\varepsilon - W + I + J,$$

where in a simple picture one can consider $\Delta\varepsilon$ to be equal to the HOMO-LUMO gap of a closed-shell molecule (hence directly related to the IP and EA through Koopman's theorem), I, J so-called excitation transfer integrals, and W the *on-site* screened Coulomb interaction.

On the other hand, a CT exciton has an energy which may be written as:

$$E_{CT} = \Delta\varepsilon - \bar{W},$$

where \bar{W} the inter-molecular screened Coulomb interaction, which in this case represents the Coulomb interaction between neighbouring donor-acceptor moieties.

Let us now consider a finite displacement δu along a given vibrational mode of frequency ω . The coupling of the excited states to the vibration is proportional to the first derivative of their energy along δu , therefore, if g is the coupling constant of each state to the mode (LE: Local exciton (non-charge transfer), CT: Charge transfer exciton):

$$g_{LE} - g_{CT} \cong \frac{\delta \bar{W}}{\delta u} - \frac{\delta W}{\delta u},$$

Where we have assumed that the change of the transfer integrals is small.

There are two cases to distinguish:

1. Low-frequency vibrational mode

In this case the displacement δu results in changes in the distance between the D-A moieties, and does not significantly alter the intramolecular structure. Therefore, $\frac{\delta W}{\delta u} \cong 0$ and $\frac{\delta \bar{W}}{\delta u}$ remains finite, hence $|g_{LE}| < |g_{CT}|$.

2. High-frequency vibrational mode

In this case the displacement δu results in changes in the internal structure of the D-A moieties, while their intermolecular distance may be considered to remain the same.

Therefore, here we have $\frac{\delta \bar{W}}{\delta u} \cong 0$ and $\frac{\delta W}{\delta u}$ remains finite, resulting in $|g_{LE}| > |g_{CT}|$.

Therefore, overall, this simple picture suggests that a local exciton is expected to couple preferentially to high-frequency vibrations, while excitons with strong charge transfer character will exhibit stronger coupling to low-frequency vibrations. This is consistent with fully first-principles results in organic molecular crystals^{21,22}.

2) In non-fullerene acceptor molecules for OPV (such as ITIC, o-IDTBR, Y6, etc), the HOMO and LUMO orbitals are delocalized on almost the whole pi-conjugation backbones. The first singlet excited states for these molecules show weak CT-excitation character. From Fig. 3c, Y6 and Y5 with emission wavelength of ~ 800 nm show similar non-radiative decay rates compared to o-IDTBR with 740 nm emission and IO-4Cl with 680 nm emission. This experimental results cannot be rationalized by the current arguments of the authors. Why? It will impact generality of their conclusions.

Response: We greatly appreciate the thoughtful feedback provided by the reviewer, and we would like to address their comments regarding a part of the study on non-fullerene acceptor molecules (NFAs).

We agree with the reviewer that non-fullerene acceptor molecules studied here have the frontier molecular orbitals which are delocalised over the whole π -conjugated backbone. As pointed out by the reviewer, NFA's $S_0 \rightarrow S_1$ transition indeed shows very weak charge-transfer character. We believe it is important to note that charge transfer character in the electronic transition of those NFA systems are notably lower with respect to the OLED emitter (radical and

conventional TADF) studied where those OLED emitter systems have weaker absorption cross-section due to weaker electron-hole overlap. Conversely, NFAs have been purposefully engineered to possess strong absorption properties tailored by stronger electron-hole overlap specifically for their application in organic photovoltaics (OPV).

The reviewer has correctly noted the similarity in nonradiative rates among NFAs with different emission wavelengths. It's important to clarify that while the reviewer suggests a potential misalignment of these results with the general conclusions outlined in our manuscript, we respectfully emphasize that these specific results were only used as supportive control for the conclusion drawn in the manuscript.

Our intension is to study the NFAs because those systems 1) neither poses strong non-bonding type orbitals for the electron and hole accepting levels, 2) nor those systems have vibrational localization induced by strong structural twisting convoluted with the disjoint electron and hole pair, which are the two key design principles proposed in the manuscript to supress strong coupling to high frequency modes. So, we wanted to study those state-of-the-art materials as control and we indeed saw strong coupling to the high-frequency modes ($>1000\text{ cm}^{-1}$) for all the NFA studied as presented in the Extended Data Fig. 2 which bolster our conclusion about those two design principles.

Nevertheless, it is worth noting that the observed non-radiative losses in multiple NFAs emitting at various wavelengths do not exactly conform to the expected trends predicted by the energy-gap law and this point deserves a more general answer. In response to this comment, our newly added section in the supplementary information highlight some key notes:

Section 21: Extended discussion on vibrational coupling and non-radiative decay in non-fullerene acceptors:

As presented in the Extended Data Fig.2, all the NFA molecules that we studied show much stronger coupling to the high-frequency ($>1000\text{ cm}^{-1}$) in comparison to the radical and TADF molecules studied. This observation can be explained as the absence of - a) pure non-bonding character in the hole and electron accepting levels, b) disjoint electron and hole wavefunction promoted by charge-transfer character and strong structural twisting.

Supplementary Fig. 31 : a, PLQE trends by the series organic chromophores and NFAs studied here plotted against emission wavelength. Data of the other chromophores are taken from ref^{23,24}. The solid black line represents the combined dataset.

We note that in NFAs, the $S_0 \rightarrow S_1$ transition exhibits weak charge-transfer character, primarily induced by push-pull effect by the heteroatoms²⁵. This push-pull type effect predominantly contribute to the reduced bandgaps in NFAs. However, this charge-transfer character in the $S_0 \rightarrow S_1$ transition of NFAs is notably weaker than for highly twisted, non-planar TADF's and the radical's lowest energy electronic transition. Consequently, radicals (such as TTM-3PCz, TTM-3NCz, TTM-TPA, M₂TTM-3PCz) and TADF's (APDC-DTPA, 4CzIPN) studied exhibit lower absorption coefficients with spatially-separated electron-hole pairs, in contrast to NFAs, which display a more substantial overlap between electron and hole wavefunctions (as illustrated in Supplementary Fig. 14) and are chemically engineered to efficiently absorb photons for use in organic photovoltaic (OPV) applications. As a result of the stronger coupling to the high-frequency modes exceeding 1000 cm^{-1} (similar to Rhodamine-6G and rr-P3HT, main), PLQE trends in NFA molecules closely adhere to the energy-gap law, similar to conventional dyes (refer to Supplementary Figure 31 for details).

Supplementary Fig. 32: Non-radiative rates plotted for the NFAs studied.

Supplementary Fig. 33: a, vibrational coherence extracted from the IVS experiment (integrated in the photo-induced absorption region), b, Corresponding $|FFT|^2$ spectra

Upon closer examination, as illustrated in Supplementary Figure 31, a noticeable and consistent disparity becomes clear across all sets of dyes and semiconducting molecules, encompassing Squaranines, PBI, BODIPY, Cyanine, PPCy dyes, and NFAs. Even when comparing data points within the same class, multiple instances exist where a lower-bandgap system exhibits nearly identical or even higher PLQEs when compared to its bluer counterpart. Similar observations can be made for the NFAs that were investigated in our work.

One key finding is the absence of a significant increase in non-radiative rates when transitioning from IO-4Cl to o-IDTBR and Y6, as depicted in Supplementary Figure 32. This phenomenon can be elucidated by variations in the chemical structures of these molecules, resulting in alterations in intrinsic vibrational frequencies and effective exciton-vibrational coupling, as illustrated in Supplementary Figure 33. Notably, the strongly coupled high-frequency modes in Y5 and Y6 have lower vibrational frequencies compared to IO-4Cl and o-IDTBR, which is the main reason for the similar non-radiative rates observed between IO-4Cl to o-IDTBR and Y6.

3) Regarding the point, i.e., reduced exciton-vibration coupling for CT exciton, the authors have examined several different kinds of organic molecules, which is sufficient to exemplify it. However, regarding the point “Non-bonding character of electron and hole levels”, the authors just examined two radical systems carefully. So, I suggest the authors examined other different molecular systems (e.g., non-fullerene acceptors for OPV, TADF molecules,, etc) to verify its generality in other systems.

Response: We thank the reviewer for the suggesting to explore the non-bonding nature of the e^- and h^+ accepting levels in other systems like OPV and TADF in more detail.

In the current manuscript, we have studied radical systems (with both non-bonding electron and hole levels), conventional TADF systems (with predominant non-bonding hole levels) and OPV materials (with π and π^* type hole and electron accepting levels). As the reviewer correctly pointed out, in the later section of the manuscript, we made deliberate efforts to modify the non-bonding character within the hole accepting level through site specific chemical adjustments in the radical emitter systems. This endeavour resulted in a notable reduction in the non-radiative decay rate. We appreciate the reviewer’s suggestion to extend the study of tuning non-bonding character in other systems like OPV and TADF.

Non-fullerene acceptor for the OPV: Interestingly, to the best of our knowledge, all the OPV materials that we are aware of (including state-of-the-art π -conjugated polymers or non-fullerene acceptors) appears to have dominant bonding (π) or antibonding (π^*) type hole or electron accepting levels and lacks pure non-bonding character. In this context we would like to acknowledge the work by Forrest and coworkers^{26,27} where they predict the peripheral -CN, -Cl group attached on some non-fullerene acceptors can introduce some non-bonding character which can be responsible for the lower reorganization energy (λ) during the course of charge transfer. We would also like to acknowledge the work by Chao and coworkers^{28,29} where they theoretically predict that, the local nonbonding character in frontier orbitals can be generated from multiple orbital interactions involving π and π^* orbitals of even alternant hydrocarbons

such as naphthalene which is also dependent of the redox state. Some of the OFETs³⁰ can poses non-bonding type orbitals just for HOMO levels in very planner molecules.

It is noteworthy that molecular orbitals in those cases discussed above are still highly dominated by π , π^* character (sometimes not purely non-bonding type), specific to certain redox state. Individually boosting nonbonding character in particular e^- and h^+ levels for such materials are chemically challenging due to the planner and highly π -conjugated structures whereas, for the radials studied (in the main text) it was possible due to their strong CT-type electronic transition ($D \rightarrow A$) and flexibility to attach (D,A) at different to manipulate the non-bonding character. However, it is important to note that synthesizing and designing an entirely new chromophore with purely non-bonding orbitals is a time-consuming process that goes beyond the scope of the current paper and is therefore suggested to the wider community.

Hence, in the concluding remarks of this paper, we propose the concept of exploring the design of the OPV materials with purely non-bonding type orbitals as follows:

“However, the suppression of non-radiative decay pathways due to the charge-transfer character of the excitations and non-bonding nature of the levels, as demonstrated here, overcomes this and enables high luminescence efficiency from these states..... **The proposed design-principles also opens up new possibilities for OPVs**, by allowing efficient radiative recombination in OPVs (such as achieved in metal-halide perovskites or GaAs solar cells) to boost Voc, the major outstanding challenge in the field^{3,13}.”

[This has been redacted]

[Redacted text block containing multiple lines of blacked-out content]

[This has been redacted]

4) In Supplementary Figure 16, HOMO and LUMO wavefunctions in triphenylamine both show non-bonding character. Why in LUMO electron-vibration coupling related to high-frequency vibrations cannot be suppressed compared to in HOMO?

Response: We agree with the reviewer that LUMO orbital has some non-bonding character especially for the connectivity (C3-C4-C5, C2'-C1'-C6', C3'-C4'-C5', C2''-C1''-C6'', C3''-C4''-C5''). We thank the reviewer for pointing this out. Now we are adding (in the supplementary information) the atom-by-atom descriptions of the orbital character for all the molecular orbitals for which the vibrational coupling calculation was performed. We observe hybridised π^* orbital mixed with some non-bonding character for the LUMO which helps us to explain its stronger coupling to the phenylic ring stretching modes with respect to the purely nb-type HOMO orbital. We added the new figure and explanatory note in the supplementary information now.

“As shown in the supplementary table 1, for the LUMO orbital there is some non-bonding character for the connectivity (C3-C4-C5, C2'-C1'-C6', C3'-C4'-C5', C2''-C1''-C6'', C3''-C4''-C5'') but at the same time there is some strong π^* -type anti-bonding character present on the phenyl ring (for the connectivity C5- C6, C2-C3, C5'- C6', C2'-C3', C5''- C6''). On the other hand all connectivity in the phenyl rings are non-bonding type on the phenyl ring with predominant coefficient on the central N atom which explains stronger coupling to the phenyl ring-stretching modes in the hybridised nb- π^* type LUMO with respect to the N-nb type HOMO. HOMO-1, HOMO-2 has purely π -bonding type character in the phenylic ring and LUMO+1 and LUMO+2 have purely π^* -anti-bonding type character and strongly couples to the phenylic ring stretching modes.

Supplementary Figure 16.2: atom-by-atom description of TPA

Supplementary Table 1: Nature of the molecular orbital of TPA (with iso-surface value 0.075)

MO	Assignment of the MO	Occupation	Connections	Connectivity
63	HOMO-2	2	C5'- C6' C2'-C3' C5''- C6'' C2''-C3''	Bonding Bonding Bonding Bonding
64	HOMO-1	2	C5- C6 C2-C3	Bonding Bonding
65	HOMO	2	N-C1-C2 N-C1-C6 C2-C3-C4 C4-C5-C6 N-C1'-C2' N-C1'-C6' C2'-C3'-C4' C4'-C5'-C6' N-C1''-C2'' N-C1''-C6'' C2''-C3''-C4''	Non-bonding Non-bonding Non-bonding Non-bonding Non-bonding Non-bonding Non-bonding Non-bonding Non-bonding Non-bonding Non-bonding

			C4''-C5''-C6''	Non-bonding
66	LUMO	0	C5- C6	Anti-bonding
			C2-C3	Anti-bonding
			C5'- C6'	Anti-bonding
			C2'-C3'	Anti-bonding
			C5''- C6''	Anti-bonding
			C2''-C3''	Anti-bonding
			C2-C1-C6	Non-bonding
			C3-C4-C5	Non-bonding
			C2'-C1'-C6'	Non-bonding
			C3'-C4'-C5'	Non-bonding
			C2''-C1''-C6''	Non-bonding
			C3''-C4''-C5''	Non-bonding
67	LUMO+1	0	C1''-C6''	Anti-bonding
			C1''-C2''	Anti-bonding
68	LUMO+2	0	C1-C6	Anti-bonding
			C1-C2	Anti-bonding
			C1'-C6'	Anti-bonding
			C1'-C2'	Anti-bonding

5) The precision for visualization of molecular orbitals and excited states should be given, since its precision will impact judgement of non-bonding character. In Section 14 in SI, the precision for visualization of molecular orbitals is not good. The authors need to unify the precision for their visualization.

Response: We thank the reviewer for their comment on the visualization of the molecular orbital and excited state wavefunctions. We are using an iso-value of 0.075 for all the MOs of carbazole and triphenylamine, which we now explicitly state in the SI in order to ensure clarity. We have therefore made the following changes:

“Supplementary Figure 16: Huang-Rhys Factor of the frontier molecular orbitals of triphenylamine (TPA). Frontier molecular are plotted with iso-value 0.075 in the right panel of the figure, at the ground state optimised geometry.”

“Supplementary Figure 17: Huang-Rhys Factor of the frontier molecular orbitals of the N-phenylcarbazole (PCz): Frontier molecular are plotted with iso-value 0.075”

We now standardized the iso-surface value at 0.075 and maintained consistency across other figures, including Supplementary Figures 19, 20, and 21. The iso-value has been updated in the respective figure captions for clarity and accuracy.

[Reference]

1. Gould, I. R., Noukakis, D., Goodman, J. L., Young, R. H. & Farid, S. A quantitative relationship between radiative and nonradiative electron transfer in radical-ion pairs. *J Am Chem Soc* **115**, 3830–3831 (1993).
2. Barbara, P. F., Meyer, T. J. & Ratner, M. A. Contemporary Issues in Electron Transfer Research. *J Phys Chem* **100**, 13148–13168 (1996).
3. Liu, Q. & Vandewal, K. Understanding and Suppressing Non-Radiative Recombination Losses in Non-Fullerene Organic Solar Cells. *Advanced Materials* vol. 35 Preprint at <https://doi.org/10.1002/adma.202302452> (2023).
4. Benduhn, J. *et al.* Intrinsic non-radiative voltage losses in fullerene-based organic solar cells. *Nat Energy* **2**, (2017).
5. Azzouzi, M. *et al.* Nonradiative Energy Losses in Bulk-Heterojunction Organic Photovoltaics. *Phys Rev X* **8**, (2018).
6. Jortner, J. Temperature dependent activation energy for electron transfer between biological molecules. *J Chem Phys* **64**, 4860–4867 (1976).
7. Wang, S. F. *et al.* Polyatomic molecules with emission quantum yields >20% enable efficient organic light-emitting diodes in the NIR(II) window. *Nat Photonics* **16**, 843–850 (2022).
8. Wei, Y. C. *et al.* Overcoming the energy gap law in near-infrared OLEDs by exciton–vibration decoupling. *Nat Photonics* **14**, 570–577 (2020).
9. Zewail, A. H. Optical molecular dephasing: principles of and probings by coherent laser spectroscopy. *Acc Chem Res* **13**, 360–368 (1980).
10. Nelson, K. A. & Williams, L. R. Femtosecond time-resolved observation of coherent molecular vibrational motion. *Phys Rev Lett* **58**, 745–745 (1987).
11. Ha, J. M. Y. *et al.* Observation of molecular vibrations in real time. *Phys Rev Lett* **57**, 3302–3302 (1986).
12. Ruhman, S., Joly, A. G. & Nelson, K. A. *Coherent Molecular Vibrational Motion Observed in the Time Domain Through Impulsive Stimulated Raman Scattering*. *IEEE JOURNAL OF QUANTUM ELECTRONICS* vol. 24 (1988).
13. Ghosh, T., Aharon, S., Etgar, L. & Ruhman, S. Free Carrier Emergence and Onset of Electron-Phonon Coupling in Methylammonium Lead Halide Perovskite Films. *J Am Chem Soc* **139**, 18262–18270 (2017).
14. Fu, J. *et al.* Electronic States Modulation by Coherent Optical Phonons in 2D Halide Perovskites. *Advanced Materials* **33**, (2021).
15. Ai, X. *et al.* Efficient radical-based light-emitting diodes with doublet emission. *Nature* **563**, 536–540 (2018).
16. Abdurahman, A. *et al.* Understanding the luminescent nature of organic radicals for efficient doublet emitters and pure-red light-emitting diodes. *Nat Mater* **19**, 1224–1229 (2020).
17. Alvertis, A. M., Schröder, F. A. Y. N. & Chin, A. W. Non-equilibrium relaxation of hot states in organic semiconductors: Impact of mode-selective excitation on charge transfer. *Journal of Chemical Physics* **151**, (2019).

18. Wang, S. F. *et al.* Polyatomic molecules with emission quantum yields >20% enable efficient organic light-emitting diodes in the NIR(II) window. *Nat Photonics* **16**, 843–850 (2022).
19. Wilson, J. S. *et al.* The energy gap law for triplet states in pt-containing conjugated polymers and monomers. *J Am Chem Soc* **123**, 9412–9417 (2001).
20. Cudazzo, P., Gatti, M. & Rubio, A. Excitons in molecular crystals from first-principles many-body perturbation theory: Picene versus pentacene. *Phys Rev B Condens Matter Mater Phys* **86**, (2012).
21. Alvertis, A. M., Haber, J. B., Engel, E. A., Sharifzadeh, S. & Neaton, J. B. Phonon-Induced Localization of Excitons in Molecular Crystals from First Principles. *Phys Rev Lett* **130**, (2023).
22. Alvertis, A. M. *et al.* Impact of exciton delocalization on exciton-vibration interactions in organic semiconductors. *Phys Rev B* **102**, (2020).
23. Vasilopoulou, M. *et al.* Advances in solution-processed near-infrared light-emitting diodes. *Nature Photonics* vol. 15 656–669 Preprint at <https://doi.org/10.1038/s41566-021-00855-2> (2021).
24. Mayerhöffer, U., Gsänger, M., Stolte, M., Fimmel, B. & Würthner, F. Synthesis and molecular properties of acceptor-substituted squaraine dyes. *Chemistry - A European Journal* **19**, 218–232 (2013).
25. Liu, W. *et al.* Low-Bandgap Non-fullerene Acceptors Enabling High-Performance Organic Solar Cells. *ACS Energy Lett* **6**, 598–608 (2021).
26. Li, Y., Huang, X., Sheriff, H. K. M. & Forrest, S. R. Semitransparent organic photovoltaics for building-integrated photovoltaic applications. *Nature Reviews Materials* vol. 8 186–201 Preprint at <https://doi.org/10.1038/s41578-022-00514-0> (2023).
27. Liu, X., Li, Y., Ding, K. & Forrest, S. Energy loss in organic photovoltaics: Nonfullerene versus fullerene acceptors. *Phys Rev Appl* **11**, (2019).
28. Chen, W. C. & Chao, I. Molecular orbital-based design of π -conjugated organic materials with small internal reorganization energy: Generation of nonbonding character in frontier orbitals. *Journal of Physical Chemistry C* **118**, 20176–20183 (2014).
29. Chang, Y. C. & Chao, I. An important key to design molecules with small internal reorganization energy: Strong nonbonding character in frontier orbitals. *Journal of Physical Chemistry Letters* **1**, 116–121 (2010).
30. Liu, W. S., Liu, C. C. & Kuo, M. Y. High-performance p-channel organic smiconducting candidates bon benzo[1,2-k;4,5-k']difluoranthene derivatives. *Chemistry - A European Journal* **15**, 5896–5900 (2009).
31. Hatakeyama, T. *et al.* Ultrapure Blue Thermally Activated Delayed Fluorescence Molecules: Efficient HOMO-LUMO Separation by the Multiple Resonance Effect. *Advanced Materials* **28**, 2777–2781 (2016).
32. Schnedermann, C. *et al.* A molecular movie of ultrafast singlet fission. *Nat Commun* **10**, (2019).

Reviewer Reports on the First Revision:

Referees' comments:

Referee #1 (Remarks to the Author):

The authors addressed all the comments raised in an extensive and convincing manner. I therefore accept the publication of this manuscript without any further request.

Referee #3 (Remarks to the Author):

The authors replied my questions carefully, although one of my comments was not addressed well. The case of non-fullerene acceptors (NFAs) cannot be rationalized by their conclusions. In the present manuscript, the discussions on NFAs mislead readers. I suggest the authors delete the results and discussions about NFAs.

Author Rebuttals to First Revision:

Review report and response to reviewers for the revised manuscript (Manuscript: 2022-09-14984B)

Referee #1 (Remarks to the Author):

The authors addressed all the comments raised in an extensive and convincing manner. I therefore accept the publication of this manuscript without any further request.

Response: We thank the reviewer for thorough evaluation of our manuscript and constructive comments, and we are pleased to know that all concerns have been comprehensively addressed. The review significantly contributed to the improvement of our work.

Referee #3 (Remarks to the Author):

The authors replied my questions carefully, although one of my comments was not addressed well. The case of non-fullerene acceptors (NFAs) cannot be rationalized by their conclusions. In the present manuscript, the discussions on NFAs mislead readers. I suggest the authors delete the results and discussions about NFAs

Response: We sincerely thank the reviewer for their thorough evaluation, and we are delighted to note that the majority of the comments have been comprehensively addressed.

Concerning the NFA, we acknowledge that NFA systems deviate from the design principles proposed in the manuscript, exhibiting a more pronounced exciton-vibration coupling at high frequencies. This characteristic aligns them more closely with conventional molecules, as outlined in the manuscript's introductory section. Consequently, we have enhanced the clarity of our presentation by revising the figure to exclude the NFA portion. Additionally, in the main text, we have streamlined the discussion on NFAs, relocating it to the Supplementary Information (SI), and provided a concise reference to it in the main text for context.